# The Impact of Differences in Retrieval Algorithms between Processing Centers on GNSS Radio Occultation Refractivity Retrievals in the Planetary Boundary Layer

Sara Vannah[1], Stephen S. Leroy[1], Chi O. Ao[2,*], E. Robert Kursinski[3,*], Kevin J. Nelson[2,*], Kuo-Nung Wang[2,*], and Feiqin Xie[4,*]

[1]Atmospheric and Environmental Research, Inc., Lexington, MA, USA 02421
[2]Jet Propulsion Laboratory, California Institute of Technology, Pasadena, CA, USA 91109
[3]PlanetiQ, 15000 West 6th Avenue, Suite 202, Golden, CO, USA 80041
[4]Department of Physical and Environmental Sciences, Texas A&M University, Corpus Christi, TX, USA 78412
[*]These authors contributed equally to this work.

**Correspondence:** Sara Vannah (svannah@aer.com)

**Abstract.** GNSS radio occultation (GNSS RO) performance in the planetary boundary layer is strongly dependent on retrieval algorithms. In this work, we characterize differences in refractivity retrievals in the planetary boundary layer across three major processing centers of GNSS RO data — NASA JPL, ROM SAF, and UCAR. Using a shared base of occultations from the FORMOSAT-3/COSMIC-1 GNSS RO mission, we identify key differences between the three processing centers that are especially strong in the regions of frequent super-refraction. We find that the minimum penetration height allowed by each processing center is correlated with the amount of super-refraction, resulting in poorer penetration and higher refractivity biases in the Tropics. We found JPL to have the most conservative minimum height in this region at 1 km, followed by ROM SAF (640 m), and UCAR (420 m). We identify two key geopotential heights — 0.8 km and 2.6 km — to sample the global distribution of inter-center refractivity bias, finding differences of 0.3–0.5% in the Tropics. We also find negative refractivity biases of up to $-4\%$ relative to ERA5 reanalysis in regions of persistent high stratocumulus coverage, and areas along the descending branch of the Hadley circulation, with negligible bias along the intertropical convergence zone. A comparison to ERA5 also reveals areas of weak (0.2–0.5%) positive refractivity biases in polar regions. We hypothesize potential causes for these biases based on truncation schemes, radio-holographic filtering choices, and quality control, and identify findings deserving of further investigation.

## 1 Introduction

Radio occultation (RO) using Global Navigation Satellite Systems (GNSS) transmission to low Earth orbit receivers provides a remote sensing capability uniquely suited to climate monitoring and atmospheric process studies due to its all-weather capability, global coverage, and extremely high vertical resolution (Kursinski et al., 1997, 2000). These capabilities have placed GNSS RO as one of the leading contributors to data assimilation systems for numerical weather prediction (Bauer et al., 2014; Cardinali and Healy, 2014) and provided the strongest evidence for multi-decadal warming of the tropical upper troposphere

(Steiner et al., 2020; Vergados et al., 2021; Gleisner et al., 2022; Johnston et al., 2018; Johnston and Xie, 2018; Verkhoglyadova et al., 2014). The high vertical resolution of GNSS RO has been useful for atmospheric process studies regarding tropopause variability (Randel et al., 2003; Randel and Wu, 2005), atmospheric gravity waves (Tsuda et al., 2000; Wang and Alexander, 2010; Schmidt et al., 2016; Alexander et al., 2024) and tropical cyclone structure (Vergados et al., 2013; Lasota et al., 2020).

Historically, numerical weather prediction and atmospheric process studies using GNSS RO have focused on the upper troposphere/lower stratosphere (UTLS), where the accuracy and precision of GNSS RO is strongest. Indeed, by virtue of its exceptional accuracy, it is assimilated into numerical weather prediction systems without variational bias correction, thereby anchoring analyses and atmospheric reanalyses (Lasota et al., 2020). In the last two decades, GNSS RO retrieval scientists have sought to extend these capabilities into the atmospheric planetary boundary layer (PBL), which is central to important
problems in atmospheric and climate modeling but has proven challenging to measure from space (National Academies of Sciences, Engineering, and Medicine, 2018; Teixeira et al., 2021). The foremost problems for RO sounding in the PBL have been known for nearly 30 years (Kursinski et al., 1997): along-track heterogeneity of water vapor, atmospheric multi-path and diffraction, super-refraction, and low signal amplitude. New developments in RO instrumentation and measurement techniques, as well as retrieval algorithms have steadily chipped away at these problems.

The most prohibitive obstacle remaining in RO sounding of the PBL are the entangled problems of weak signals and super-refraction. The problem of super-refraction is based in the physics of the RO observation. Vertical gradients in the microwave index of refraction can become so sharp that rays with tangent points within the ducting layer have a radius of curvature smaller than the radius of the Earth. This causes total internal reflection inside the ducting layer, rendering the ducting layer invisible to external rays: a phenomenon known as super-refraction (Sokolovskiy, 2003; Sokolovskiy et al., 2014, 2024). There is no
single solution to retrieve a refractivity profile with super-refraction present from a bending angle profile (Xie, 2006). While RO performance remains unaffected above the ducting layers, the "invisibility" of the duct results in large negative biases in retrieved refractivity and even larger fractional biases in retrieved water vapor below the duct (Xie, 2006). Super-refraction is closely correlated with the presence of marine stratocumulus (Xie et al., 2010). A general approach to mitigating the negative biases below super-refraction ducts by using additional external information to correct the super-refraction impact height was
suggested by (Xie, 2006). Examples of external information in proposed unbiased retrievals include a measurement of total column water vapor (Wang et al., 2017), the synchronously received signal reflected off the ocean surface (Wang et al., 2020), and collocated nadir radiance microwave data (Wang et al., 2024). Furthermore, all of these methods rely on robust tracking of the RO signal deep into the region where the straight-line tangent point falls below Earth's surface. This is complicated for two reasons: strongly bent signals are also strongly defocused; and that uncertainties in models of the total optical path delay
of such signals begets further signal loss where demodulating satellite-identifying pseudo-random codes and navigation data messages, standard in the GNSS RO retrieval (Sokolovskiy, 2001, 2003). GNSS RO inversion also relies on an assumption of local spherical symmetry — that atmospheric temperature, pressure, and water vapor depend only on height in the vicinity of an RO sounding — for retrieval, but this assumption is strongly violated by water vapor morphology in the PBL, especially in low latitudes. No evidence, however, has suggested that RO does not capture a horizontal path-average retrieval of water
vapor for any given sounding. Therefore, it is widely assumed that RO sounding in the PBL can be applicable to research

that does not require horizontal resolution less than approximately 70 km, such as climate monitoring. Wave optics retrieval algorithms for RO based in Fourier integral operators (Gorbunov et al., 2004) have innovated performance in the PBL by almost entirely eliminating difficulties associated with atmospheric multi-path and Fresnel diffraction. These approaches treat RO signal propagation as a problem of wave physics rather than geometric optics, as though RO were a problem of holography wherein the GNSS electric wave field can be reconstructed everywhere, even in post-processing.

The most recent, generation 3 GNSS RO missions have deployed instruments that track GNSS RO signals with much greater gain than previous generations of RO instruments. The proof-of-concept generation 1 GPS/MET mission obtained RO measurements with a signal-to-noise (SNR) ratio of approximately 300 V/V (1 Hz). The generation 2 missions, including CHAMP, SAC-C, Metop-A,B,C, and COSMIC-1, obtained median SNRs of roughly 700 V/V (1 Hz). The generation 3 missions, including the six satellites of COSMIC-2 and the satellites of the commercial RO provider PlanetiQ, obtain median SNRs of roughly 1500 V/V (1 Hz) (Schreiner et al., 2020). Gorbunov et al. (2022a) introduced two forms of normalized SNR to assess the noise floor of each individual profile, showing that a higher SNR leads to better penetration in the PBL. Gorbunov et al. (2022b) confirmed that COSMIC-2 has a better noise floor than older missions. These new, exceptionally large SNRs allow deeper signal tracking than ever before, even in the presence of extreme bending and super-refraction, reducing refractivity biases.

Very recent results show that these high-SNR RO soundings enable the detection of the presence of super-refraction (Sokolovskiy et al., 2024) and the critical refractional radius[1] that defines the super-refraction duct, so long as the bottom of the ducting layer is elevated above Earth's surface (Zeng et al., 2024). The duct refractional radius is a key parameter in algorithms to mitigate the negative biases induced by super-refraction. These biases have a strong impact on water vapor retrievals (Ho et al., 2010). UCAR has recently begun publishing level 1 (calibrated excess phase) COSMIC-2 RO data with a super-refraction detection flag and a value for the duct refractional radius; those data can then be used in a retrieval of water vapor from RO data that seeks to be unbiased.

While unbiased retrievals of water vapor in the PBL from RO data have yet to be published, systematic and structural differences in existing retrievals may be identified. The goal of this work is to characterize these differences in the PBL, so that their root causes may determined in future study. The components of existing retrieval algorithms that can induce bias include implementations of navigation message demodulation (Sokolovskiy et al., 2006), radio-holographic filtering in wave optics retrievals and the smoothing it begets (Sokolovskiy et al., 2010), and the approach to cutting off an RO signal low in the atmosphere when the signal becomes too weak to be of use (Sokolovskiy et al., 2010). First, GNSS signals are all modulated by prescribed pseudo-random codes at rates of 1.023 MHz for the GPS L1 C/A code and 10.23 MHz for the GPS L1 and L2 P codes. This sequence of bit flips (or $180°$ phase shifts) is typically removed by the RO instrument itself in the course of tracking. Many GNSS signals also transmit navigation data as another, much slower modulation: 50 Hz in the case of the Global Positioning System (GPS). That navigation message bit stream is data, meaning it is not fixed from one epoch to the next, and so it cannot be removed by the RO receiver when tracking a signal in an "open-loop" mode (Sokolovskiy, 2001). Those bit streams are collected by ground GNSS stations, and their associated $180°$ phase flips are removed in post-processing. Complete removal of the navigation bit stream relies upon precise time matching, which will never be perfectly precise due

---

[1]The "refractional radius" for a parcel of air is defined as its distance to the RO center of curvature multiplied by its microwave index of refraction.

to atmospheric influence on the RO signal. This imprecision causes signal loss and phase noise in RO retrieval in the PBL. Secondly, radio-holographic filtering is implemented in retrieval systems in order to isolate a signal from electronic noise. These techniques apply an arbitrary bandpass filter to the Fourier transformed signal to isolate a single tone of the signal. The narrower the filter, the more precisely the signal frequency is determined; however, this will result in in more vertical smoothing in the retrieval. The wider the filter, the less precise the signal frequency, but a higher vertical resolution will result. Any RO retrieval system with a wave optics step has a radio-holographic filter with an *ad hoc* width. Third, different retrieval systems have implemented different types of wave optics retrievals. The Fourier integral operator retrievals most commonly utilized to apply this filtering are the Full Spectrum Inversion (FSI) (Jensen et al., 2003), the type 2 canonical transform (CT2) (Gorbunov and Lauritsen, 2004), and phase matching (PM) (Jensen et al., 2004). Their performances are similar but not identical (Gorbunov and Lauritsen, 2004; Sievert et al., 2018). Fourth, because RO retrieval is based on signal phase, which can only be calculated when only noise is measured, each retrieval algorithm must make an *ad hoc* decision as to the minimum height when an RO measurement has to be terminated. The implementation of this minimum height choice limits the depth of penetration of an RO retrieval, usually cutting off a retrieval a few hundred meters above the surface. This cutoff is performed in impact parameter space, and is distinct from the truncation in the time series of the signal that may be performed at an earlier step in signal processing. All of these steps will be necessary components of an RO retrieval in the PBL, and they are all associated with tunable parameters and *ad hoc* decisions.

A theoretical study of systematic error in RO retrieval in the PBL has already pointed toward two dominant structural errors in addition to super-refraction errors: the approach to determining a minimum penetration height for the signal; and a bias incurred by the addition of unbiased, random noise to a nonlinear retrieval algorithm (Sokolovskiy et al., 2010). The former is introduced above. The latter, however, is a secondary consequence of radio-holographic filtering. Each individual tone revealed in a sliding Fourier transform has a finite width, caused in part by non-stationarity of its frequency and in part by electronic white noise. The width of the filter is chosen so as to capture the real non-stationarity of the signal and to suppress noise. Some noise does leak through regardless. Because the tone's frequency drift has second- and higher-order dependencies in time, a bias in an inferred frequency results, with the same sign as the curvature of the tone's frequency in time. For most RO measurements, the sign is positive in bending angle, and the retrieved refractivity (and water therefore vapor) inherits a positive bias.

In this paper we will inter-compare RO retrieval performance in the PBL by three independent retrieval centers to characterize retrieval uncertainty. A set of studies like this has been performed in the UTLS for RO as a climate monitoring technique, collectively known as the *ROTrends* project (Ho et al., 2009, 2012; Steiner et al., 2013). With this effort we commence an effort similar to *ROTrends* but specific to the PBL. Ultimately, RO retrieves water vapor in the PBL (Kursinski et al., 1995), but with this effort we only examine retrievals of microwave refractivity as a function of geopotential height. In the Section 2 of this manuscript, we describe the RO data and models used in our analysis and give brief descriptions of the independent retrieval algorithms being compared. In the Section 3, we present our center inter-comparison analysis as well as comparing to reanalysis. Finally, in Section 4, we present our conclusions and discuss possible future directions for a larger RO-PBL inter-comparison project.

## 2 Model and data

We characterize and quantify error in GNSS RO retrieval in the PBL by intercomparing independent retrievals. Retrievals from different processing centers were assessed from the AWS Registry of Open Data using the Python application programming interface *awsgnssroutils* (Leroy and McVey, 2023; Leroy et al., 2024). We utilized the Level 2A *refractivityRetrieval* files, which contain bending angle and retrieved refractivity profiles. We filter to include only occultations from the FORMOSAT-3/COSMIC-1 mission (Rocken et al., 2000), as refractivity retrievals for this mission were generated by all three of the processing centers considered: NASA Jet Propulsion Laboratory (JPL), the COSMIC Program Office of the University Corporation for Atmospheric Research (UCAR), and the Radio Occultation Meteorology Satellite Application Facility (ROM SAF). The retrieval versions used were the 2021 UCAR reprocessing (UCAR, 2022), JPL's reprocessing version 2.6, and the 2019 processing in the Climate Data Repository (CDR) from ROM SAF (ROM SAF, 2019). This mission presented a massive increase in performance in the troposphere thanks principally to implementation of high-gain RO antennae, electronic signal amplification, and of open-loop signal tracking down to straight-line tangent altitudes of $-100$ km on the L1 channel (Liou et al., 2007). This technique uses a model to supplement sampling of the raw RO signal rather than extrapolating the rays forward in time as was done in the previously used phase-locked loop (closed loop) tracking technique (Sokolovskiy, 2001). This innovation resulted in tracking far deeper into the atmosphere than its predecessors: COSMIC-1 had a median tracking depth below 1 km, compared to approximately 3.5 km for the CHAMP mission (Anthes et al., 2008; Chang et al., 2022). For the first time, this allowed RO to probe into the PBL. The mission was limited in the PBL by a weaker signal-to-noise ratio than future missions: COSMIC-1's average signal-to-noise from 60-80 km is 800 V/V at 1Hz (compared to over 1500V/V for COSMIC-2) (Schreiner et al., 2020), which resulted in higher bending angle error from higher thermal noise. These errors propagate into errors in refractivity retrieved from the RO signal.

We additionally filtered to include only COSMIC-1 occultations processed during 2008: this provided a balance between making use of the highest yield early in the mission, while also surpassing the 17 months after its April 2006 launch to allow the constellation of six low Earth orbiter satellites (LEOs) to spread into its final orbital arrangement (Anthes et al., 2008). The contributing centers' retrievals for COSMIC-1 can be considered mature at this point. These constraints provide a database more than 2000 occultations per day, with slightly fewer (1700-2000) resulting in refractivity retrievals for each processing center. With the exception of the throughput analysis in Section 3.1, we analyze only the subset of data processed by all three centers in order to inter-compare structural differences in RO processing across centers.

Refractivity retrievals from RO follow three primary steps. The first processing step calibrates phase measurements so as to remove the influences of the transmitter's and receiver's clock biases, the satellites' motions, and the modulation of the navigation message bit-streams on the L1 C/A signal. This processing step is shared by UCAR and ROM SAF, while JPL generates excess calibrated phase information independently. The second step retrieves the bending angle of an RO ray as function of impact height (for a more comprehensive review see, e.g. Mannucci et al. (2020)). This step is performed independently by all three processing centers. Finally, refractive index ($n$) is retrieved from bending angle ($a$), as a function of impact parameter $\alpha$. In an atmosphere with little horizontal structure that satisfies the RO retrieval approximation of "local

spherical symmetry", this is done using the following Abelian integral (Hajj et al., 2002; Fjeldbo and Eshleman, 1968; Fjeldbo et al., 1971; Kursinski et al., 1996, 1997, 2000):

$$n(a) = \exp\left[\frac{1}{\pi}\int_a^\infty \frac{\alpha(a')da'}{\sqrt{a'^2 - a^2}}\right].$$ (1)

Refractivity is related to refractive index by $N \equiv (n-1)\times10^6$. Qualitatively, Eq. 1 determines the refractivity in the atmosphere by unwrapping the bending (due to atmospheric refraction) that each layer in the atmosphere contributes to total bending $\alpha$ for a single value $a$ of the ray's asymptotic miss distance from the center of curvature of the Earth as apparent at either satellite. While, in principle, Eq. 1 is integrated up to an impact parameter of infinity, in reality each processing center must make a choice of what maximum altitude to integrate up to and handle contributions due to layers higher in the atmosphere with great care. Typically, this is somewhere in the thermosphere.

In atmospheric regions that see strong vertical gradients, such as near the PBL top, a geometric optics retrieval approach fails due to atmospheric multi-path, Fresnel diffraction, and potentially super-refraction. Atmospheric multi-path occurs when multiple rays are simultaneous solutions to the geometric optics integral of Eq. 1, which occurs in general for profiles $\alpha(a)$ for which $d\alpha/da > 1/D_{\mathrm{tnm}} + 1/D_{\mathrm{rcv}}$, with $D_{\mathrm{tnm}}$ and $D_{\mathrm{rcv}}$ as the distances of the transmitter and receiver satellites to the Earth's limb, respectively. In these situations, a wave optics retrieval approach becomes necessary, because such an approach can distinguish between rays in atmospheric multi-path situations and almost completely eliminate the ringing effects of Fresnel diffraction as seen in the amplitude and the phase of the measured signal. Wave optics retrievals are based on the concept of "stationary phase" in optics (Born and Wolf, 1980). The highly efficient Fourier integral operator versions of wave optics are based in fast Fourier transforms (Gorbunov and Lauritsen, 2004; Gorbunov et al., 2004) while the "phase matching" approach is more precise (Jensen et al., 2004) yet much more computationally expensive than the Fourier integral operators. In these scenarios, radio-holographic filtering techniques are applied in order to reduce the influence of noise in a non-linear retrieval algorithm. UCAR uses a phase matching technique (Kuo et al., 2004; Sokolovskiy, 2001, 2003; Sokolovskiy et al., 2010), while JPL (Hajj et al., 2002) and ROM SAF use a type-2 canonical transform (Schwärz et al., 2024; Syndergaard et al., 2020, 2021). As explained in Section 1, Sokolovskiy et al. (2010) hypothesized that wider radio-holographic filter widths result in a larger (more positive) refractivity retrieval. This effect is especially strong around a height of 2–3 km. We have calculated geopotential height by dividing the geopotential values in the *refractivityRetrieval* files by the WMO standard value for gravity, $g_0 = 9.80665$ J kg$^{-1}$ m$^{-1}$. This implies that differences in radio-holographic filter widths between processing centers would then result in systematic refractivity biases between centers which are especially strong in regions of high super-refraction, especially in the subsiding regions of the Subtropics where marine stratocumulus is prevalent in the PBL. Centers with narrow filters are expected to have lower retrieved refractivity in the PBL.

Furthermore, quality control and choice of minimum height create retrieval differences between processing centers. ROM SAF quality control screens out retrievals with impact parameter noise above a certain threshold, as well as rejecting retrievals with bending angle or refractivity values sufficiently divergent from ionospheric correction and climate models. UCAR similarly screens out retrievals with bending angle or refractivity values that sufficiently diverge from climate models, as those with

a high standard deviation of relative difference in bending angle or refractivity with the climate model. UCAR furthermore rejects occultations with sufficiently high differences in bending angle between the L1 and L2 channels. JPL quality control rejects occultations with negative bending angles, as well as high-altitude bending angles sufficiently divergent from a fit that is exponential with height. Similar to ROM SAF and UCAR, JPL also rejects retrievals with refractivity values that significantly disagree with a climatology model, at any height. JPL is distinct from the other two centers in that it also screens retrievals with temperatures sufficiently different from the climate model. For more a more detailed comparison of quality control between the three processing centers, see Ho et al. (2012) and Sokolovskiy (2021).

Differences in ionospheric corrections across processing centers may also introduce biases. All three of the processing centers use a linear combination of L1 and L2 channel bending angles to perform their ionospheric corrections (Vorob'ev and Krasil'nikova, 1994) (ROM SAF uses an optimized combination from Gorbunov (2002c)), while extrapolating the correction terms downward lower in the atmosphere. JPL makes the transition between linear combination and extrapolation at 10 km, while UCAR and ROM SAF set the transition to extrapolation at a quality control height defined dynamically on L2 data(Ho et al., 2012). However, we anticipate the impact of differences in ionospheric correction to be a secondary impact in the PBL: Zeng et al. (2016) found that varying the height of this transition between 0-25 km caused a refractivity bias of, at most, approximately 5% in extreme cases (e.g. extrapolation transition heights below 10 km), and, on average, roughly a tenth of a percent. The average biases due to ionospheric correction are therefore considerably smaller than the retrieval biases identified in this work.

In addition to comparing between processing centers, we also assess the bias in RO retrievals relative to an atmospheric renalaysis. We compare refractivity retrieved from RO to refractivity from ECMWF's ERA5 model level hourly global reanalysis (European Centre for Medium-Range Weather Forecasts, 2022; Hersbach et al., 2020). This model refractivity was calculated using pressure ($P$), specific humidity ($Q$), and temperature ($T$) accessed from the Copernicus Research Data Archive of the UCAR (European Centre for Medium-Range Weather Forecasts, 2022). ERA5 refractivity in the microwave regime was calculated using the Smith and Weintraub (1953) formulation for refractivity $N$ as

$$N = (77.6 \text{ K hPa}^{-1})\frac{P}{T} + (3.73 \times 10^5 \text{ K}^2 \text{ hPa}^{-1})\frac{e}{T^2}, \tag{2}$$

for $e$ the water vapor partial pressure

$$e = \frac{Q}{Q + \epsilon(1-Q)}P, \tag{3}$$

where $\epsilon = 0.622$ is the ratio of the molar mass of water vapor to the molar mass of dry air. We note that ERA5 assimilated COSMIC-1 bending angle data processed by UCAR for 2006–2014, amongst other RO missions. This creates a small correlation between RO retrievals, especially from UCAR, and the reanalysis. We anticipate this correlation having little impact on our results, as quality control in data assimilation typically prevents radio occultation data from being assimilated below super-refractive layers (Cucurull, 2023).

We consider three different types of bias in refractivity retrievals: sampling bias, penetration sampling bias, and structural bias. Sampling bias is the refractivity bias caused by disparities in throughput across processing centers. This bias is considered

in Section 3.1. All results after Section 3.1, however, remove sampling bias by considering only occultations that resulted in refractivity retrievals common to all three processing centers. Penetration sampling bias is a PBL-specific sampling bias that results from the differing choice in minimum height for occultations across different processing centers. This bias is discussed at length in Section 3.2. Structural bias is the bias in a vertical refractivity profile that remains when sampling and penetration sampling bias are both removed. This represents the "true" bias in retrieved refractivity between processing centers as a result of differences in the computation of refractivity across processing centers, such as the width of the holographic filter. The total bias is computed by comparing mean refractivity bias across processing centers; sampling bias is removed by using the mean of only occultations processed by all three processing centers: this reveals structural bias combined with penetration sampling bias. Structural bias can be isolated by taking the mean bias of only occultations processed down to a particular geopotential height by all three processing centers.

We note a source of error in the RO-ERA5 comparisons: Each processing center defines the location of their reference tangent point differently. The ERA5 profile selected for each occultation was chosen to be the profile closest to the UCAR reference tangent point. Since the retrievals for each center were still performed on the same occultation — the differing tangent points are due only to differing definitions — they should all be compared to the same ERA5 profile. Using the ERA5 profile closest to, say, the ROM SAF definition of the reference tangent point may alter the RO-ERA5 bias, but should not change the bias between the processing centers themselves.

## 3 Results

### 3.1 Quality control: Overlap of Retrievals by Center

We first investigate the relative portion of occultations that result in refractivity retrievals. Differences in quality control between different processing centers at different stages — from the calibrated phase vs. time (level 1a data), to bending angle vs. impact parameter (level lb), to refractivity retrieval vs. geopotential height (level 2) — each step in processing includes checks to discard anomalous occultations. These checks may include, among other strategies, comparison of retrievals with models or using noise thresholds. Fig. 1 shows the overlap of COSMIC-1 occultations for the entire year of 2008 that result in refractivity retrievals for each processing center. The gray center region in the center of the figure shows that 67.2% of the occultations were processed by all three centers. UCAR processed the most, with 92.7% of the retrievals, followed by ROM SAF and JPL with 86.6% and 80.2% of occultations, respectively. The higher processing rate of UCAR may be a result of COSMIC-1 being a UCAR-affiliated mission. ROM SAF uses UCAR calibrated phase data for their retrievals, which may explain why they share a higher percentage of retrievals not processed by JPL (15.4%). This may indicate a difference in quality control between UCAR and JPL at the calibrated phase level. However, we note that the calibrated phase files used by the ROM SAF retrievals hosted on the AWS Registry of Open Data are an older reprocessing than those used by the UCAR retrievals. Only 2% of occultations are processed by JPL but not UCAR nor ROM SAF. These retrievals do not have any clear connection: They do not share a transmitting or receiving satellite, are a mix of rising and falling, and have a wide temporal and spatial spread. The

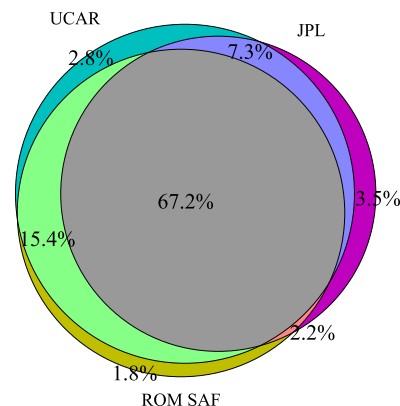

Figure 1. Top: Venn diagram illustrating the refractivity retrievals processed by each center. Lime green (15.4%), pink (2.2%), and and blue (7.3%) slices represent the overlap between UCAR and ROM SAF, ROM SAF and JPL, and JPL and UCAR, respectively. The gray center area illustrates the portion of retrievals shared by all three centers. Note that the relative size of the overlapping slices is not necessarily proportional to the relative fraction of the number of retrievals in that slice, as it not geometrically possible to have both the full circles for each center and the overlapping slices for each center be proportional in size. Bottom: Table showing the total number of rising and setting occultations for each processing center, and their intersections.

vast majority of retrievals that pass quality control for one processing center also pass for at least one of the other two centers, confirming a similar investigation by Ho et al. (2012).

## 3.2 Minimum height of penetration

The minimum height allowed by retrievals is a critical characteristic of quality control in RO processing. As a ray penetrates deep into the atmosphere, effects such as super-refraction layers, atmospheric multi-path, topography, and code demodulation all cause a decrease in SNR. Retrieval algorithms must therefore make a choice to truncate the occultation at some minimum altitude. The choice of minimum height varies by processing center as it requires balancing two effects: a lower (less conservative) minimum height can strengthen the impact of ducting and lower the precision of a retrieval, while, as explained in the introduction, a higher (more conservative) minimum height may induce a negative bias in refractivity. Each processing center makes different choices about the parameters determining how deep an occultation is allowed to penetrate. JPL minimum heights are set by fitting a single-step step function to the canonical transform amplitude, truncating at the step boundary (Ao

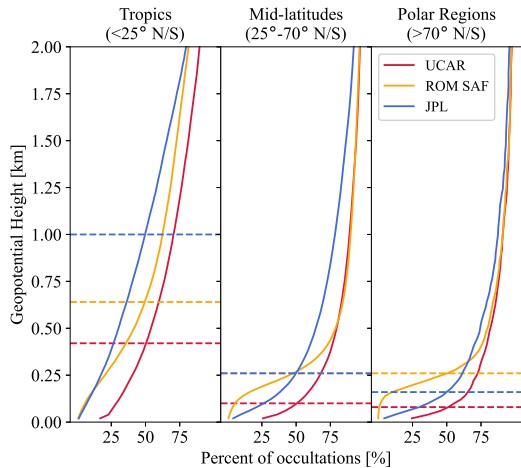

**Figure 2.** Sampling profiles (cumulative distribution functions) of minimum geopotential height of maritime occultations in winter 2008 (December 2007–February 2008) for each center, separated by latitude. Lines indicate percent of occultations, including only maritime occultations that reach below 5 km for each processing center. The solid lines indicate the percent of occultations that reach a particular altitude, while dashed horizontal lines show the altitude to which 50% of occultations penetrate for each processing center. This therefore represents the median minimum penetration height. Note these solid lines overlap for ROM SAF and JPL in the mid-latitudes (center, where the blue dashed line lies on top of the yellow) because their median minimum heights are the same to within the histogram resolution of 20 m. These occultations were not interpolated so that the center-reported minimum geopotential height could be used.

et al., 2012). UCAR determines minimum height separately for open and closed loop data: closed loop minimum heights are determined using a threshold for the difference in the filtered L1 Doppler shift and that of a model; closed loop minimum heights are determined from SNR (Sokolovskiy, 2021; Sokolovskiy et al., 2010). ROM SAF truncates when the L1 amplitude is too weak, or when the smoothed bending angle near the surface is larger 0.1 rad (Syndergaard et al., 2020).

To inter-compare the minimum allowed heights by each center, we consider only 67.2% of occultations processed by all three centers shown in Fig. 1. To isolate our analysis to PBL atmospheric effects, we restrict our analysis to include only occultations over the ocean, removing the impact of orography. We also include only retrievals that penetrate below 5 km for all three centers, reducing the impact of quality control higher in the atmosphere. The resulting distribution of minimum heights for each processing center are shown in Fig. 2.

The leftmost panel illustrates the impact of the moist Tropics on quality control: all three processing centers have significantly higher (worse) minimum height in the Tropics than at higher latitudes to account for increased uncertainty in the presence of a strong refractivity gradient. This agrees with similar results found by Ao et al. (2012). JPL (blue) is most sensitive to this effect, with only 50% of retrievals reaching 1 km. UCAR (red) has the least conservative truncation of the three centers. This is particularly evident in the Tropics, where the median minimum height of UCAR is more than 500 m deeper than that of JPL, but remains true across all latitudes. Anthes et al. (2008) found in early COSMIC-1 results that approximately 70% of

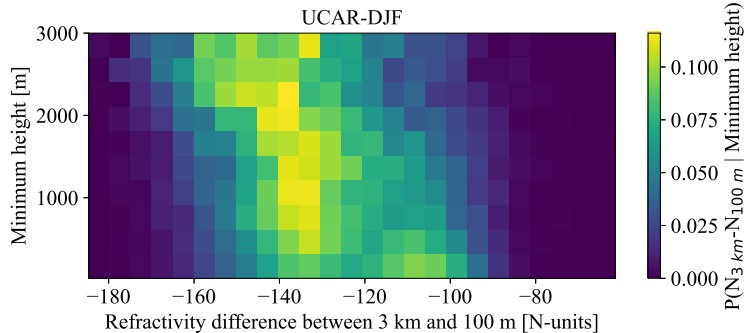

**Figure 3.** Conditional probability of refractivity difference given a particular minimum height for UCAR maritime occultations in winter (DJF) of 2008. We use refractivity difference between 3 km and 100 m as a proxy for refractivity gradient in in the PBL, as a large difference between these two heights implies a sharp gradient somewhere in the PBL that would create a super-refractive layer. Yellower coloring indicates that retrievals are more likely to fall in a particular bin. Note the negatively sloping tilt of refractivity gradient with minimum height.

ocean-only occultations penetrated below 1 km in the Tropics; this roughly agrees with our observations for UCAR and ROM SAF.

Differences in minimum height induce a sampling bias that is unique to the PBL. As our analysis considers only the subset of occultations processed by all three processing centers, a traditional sampling bias induced by differences in global coverage across soundings does not impact our results. Another sampling bias is introduced, however, by differing algorithms for determining the minimum penetration height of a retrieval. This means that a global PBL analysis — especially deep into the PBL, as at 0.8 km — will have a bias towards fewer occultations in low-latitude regions. The result is a form of sampling bias that

appears only low in the atmosphere. We dub this form of sampling bias "penetration sampling bias". This effect is separate from structural bias, the "true" bias in the vertical structure of retrievals. As our work compares only refractivity retrievals, this penetration cutoff is the combination of the cutoffs at all previous steps in processing.

    This penetration sampling bias appears strongest in regions of strong bending and super-refraction, as illustrated in Fig. 3. We treat the refractivity difference between 0.1 km and 3 km as a proxy for strong bending and the possible presence super-

refraction, as strong refractivity gradients are associated with strong bending and a super-refractive layer. This probability was computed using a two-dimensional joint probability density in refractivity difference between 0.1 km and 3 km and minimum height of maritime occultations for winter 2008 (DJF). We normalized the joint probability distribution by the marginal probability distribution in minimum height to yield the conditional probability for a refractivity different across the PBL given a specific minimum height. The highest probability region follows a negative sloping line from the bottom right

corner near $-100$ $N$-units to the upper-left near $-160$ $N$-units. This indicates that areas of larger refractivity gradient (left) are biased towards poorer penetration, enhancing the strength of the negative $N$ bias in these areas. Summer 2008 (JJA) shows similar results (not shown). The finer zonal and meridional structure of the minimum height is explored in Appendix A.

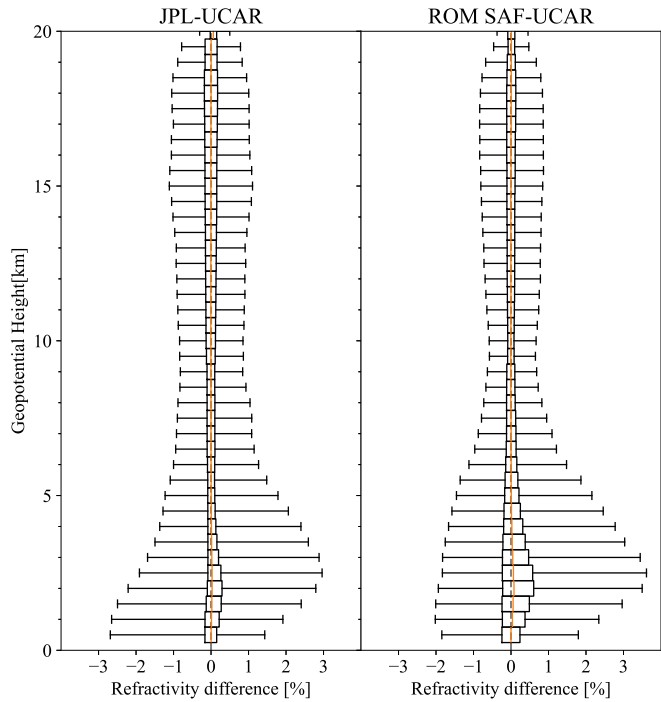

**Figure 4.** Box-and-whiskers plot showing the fractional refractivity differences between JPL and UCAR (left) and ROM SAF and UCAR (right) maritime occultations for winter (DJF) 2008. In order to compare across occultations, both traditional and penetration sampling bias are removed; this plot shows only structural biases between centers. Distributions are subset into 500 m geopotential height bins. Boxes show outline the central two quartiles of the distribution at each geopotential height while whiskers show the 1–99% ranges; orange lines show median at of each distribution. Gray dashed line indicates no refractivity difference.

## 3.3 Inter-center differences in refractivity

Differences in processing routines become clear when comparing the refractivity retrievals for different processing centers.
Fig. 4 shows the distribution of refractivity differences between processing centers. We compare both JPL and ROM SAF to UCAR, using an interpolation (without extrapolation) linear in geopotential height on a 100-m isohypsic grid between 0 and 20 km to compare occultations at different processing centers. This removes sampling and penetration sampling bias, revealing only the inter-center structural bias. While the wide flier arms dominate the plot, we find that inter-center refractivity differences for the vast majority of occultations are less than $\pm 0.15\%$. Both the spread and the median of these differences are especially 310 small from 7–10 km, and appear larger between JPL and UCAR (left) then ROM SAF and UCAR (right). These differences become larger lower in the PBL, where the ROM SAF-UCAR bias becomes larger and more positive than the JPL-UCAR bias.

To identify systematic trends in inter-center differences, we consider mean inter-center refractivity differences. The rightmost panel of Fig. 5 shows the mean difference in refractivity between JPL and UCAR (top) and between ROM SAF and UCAR (bottom), separated by latitude band. These represent structural inter-center biases. While the high latitude (purple and red) bins

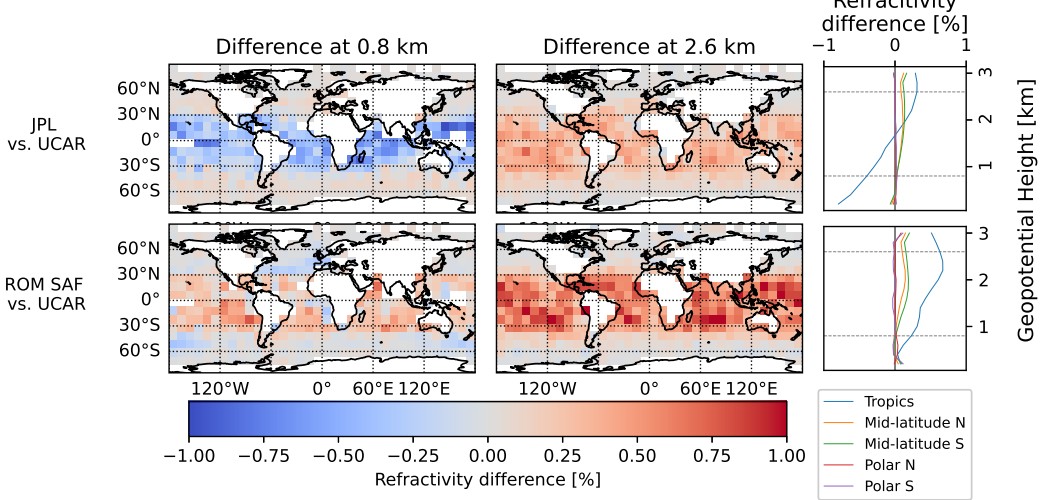

**Figure 5.** Fractional refractivity differences between JPL and UCAR (top) and ROM SAF and UCAR (bottom) for winter (DJF) of 2008. $10° \times 10°$ maps of maritime fractional refractivity differences at selected heights of 0.8 km (left) and 2.6 km (center) show the spatial variation of the distinct refractivity features found in mean refractivity difference profiles (right) indicated by gray, horizontal lines. White cells indicated masking. Note that bins which overlap the coastline still represent the median of only occultations over ocean. Many of these bins therefore contain fewer samples and are more susceptible to noise than bins whose areas are entirely over the ocean.

show near zero bias differences begin to appear in the mid-latitudes and strengthen into the Tropics. This and all subsequent plots are interpolated on a 20m geopotential height grid to asses bias at specific heights in the PBL. Both plots show positive biases relative to UCAR of 0.25–0.75% at the top of the plot. We identify a height of interest at 2.6 km to investigate this positive bias, labeled with a gray dashed line. A second area of interest is the strong bias in JPL below approximately 2 km in the Tropics, which increases in strength lower in the PBL. We choose 0.8 km as height of interest to investigate this bias,

marked with another gray dashed line. These two heights of interest are used for analysis of the global variation of bias in the left to panels of Fig. 5, and in the following sections.

      To ensure robust statistics, we bin the retrievals in Fig. 5 into $10° \times 10°$ bins, the minimum size required to ensure that every bin contains at least 30 occultations (a standard accepted minimum for Guassian statistics). To isolate the structural bias in retrievals, we consider only occultations processed down to the key geopotential heights for all three processing centers. The

left two panels show that UCAR's refractivity values are between those of JPL and ROM SAF, with a strong bias appearing in the Tropics that flips signs (and becomes weaker) at higher latitudes. Similarly, the vertical profile of the rightmost panel shows that the positive bias of ROM SAF is very large in the Tropics at 2.6 km, small but positive at 0.8 km, and near zero outside the Tropics at both heights. We note that, given the strength of the ROM SAF positive bias at 2.6 km in the Tropics, the positive bias at 0.8 km may be more indicative of strength of the positive bias at higher geopotential heights than of true

retrieval differences at that height.

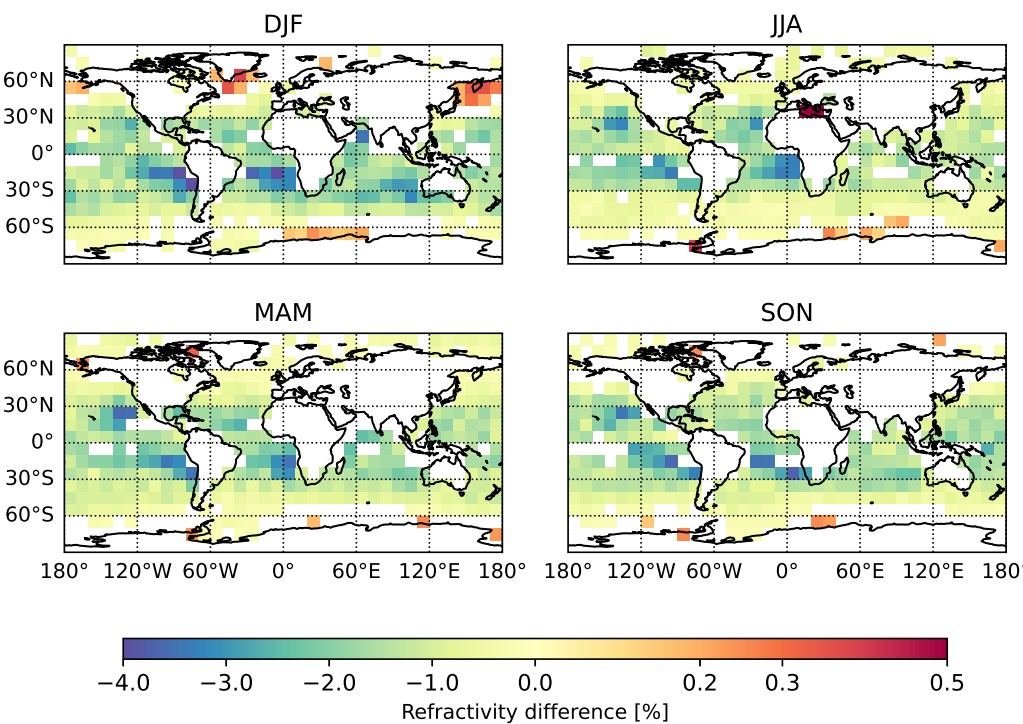

**Figure 6.** Bias between refractivity profiles retrieved from COSMIC-1 by UCAR vs. those modeled by ERA5 for maritime occultations for each season of 2008. Differences between GNSS RO, linearly interpolated in 20 m increments, and the model have been binned into $10° × 10°$ bins, chosen such that every bin contains at least 30 occultations. The maps have further been masked so that only bins with means greater than twice the standard error in the bin are shown. The color scale is set to a two-slope norm: yellow indicates 0 bias, while the differing slopes of the negative (blue) and positive (red) color scale allow the range of the negative bias strength to be shown while also making the weaker positive bias regions visible. Analogous comparisons between JPL and ERA5 and between ROM SAF and ERA5 are shown in Appendix B.

The center panel (bias at 2.6 km) shows that both ROM SAF and JPL have a positive bias relative to UCAR, again with the highest magnitude in the Tropics. ROM SAF has the larger positive bias of the two centers. This may indicate that UCAR employs a narrower radio-holographic filter, while ROM SAF utilizes a wider filter in the PBL.

### 3.4 Comparison with models

In this section, we compare GNSS RO processing in the PBL to the ERA5 reanalysis to assess bias shared across processing centers. We find that GNSS RO has a negative bias relative to ERA5 in many locations around the globe, particular in low-latitudes. The regions of strongest negative bias match those found by Xie et al. (2010); Feng et al. (2020). Comparison with

International Satellite Cloud Climatology Project (ISCPP) (Golea et al., 2016) cloud data reveals that these high negative bias regions correspond to regions of high stratocumulus coverage. Using this as a marker for moisture supports the hypothesis that the bias is due to super-refraction.

Fig. 6 shows the JPL bias relative to ERA5 for each season of 2008. Spots of strong negative refractivity bias — sometimes in excess of -4% — appear on the southwest coast of Africa and off the coast of Chile and Peru. These two were also the strongest hot spots found in similar analysis by Xie et al. (2010), who considered January-only at the same geopotential height (0.8 km), when comparing to ECMWF's ERA-Interim reanalysis (Dee et al., 2011). This demonstrates the veracity of both results. Fig. 6 also shows areas of weak positive bias at 0.8 km in high latitudes. Similar positive biases were found at 2 km by Feng et al. (2020). They appear to be strongest in the winter hemisphere, with strong bias regions near Greenland and Russia's Kamchatka Peninsula in DJF, while the bias in the Antarctic region is strongest in JJA. The seasonality indicates that the bias is unlikely due to melting sea ice. Furthermore, while these regions may feature reflections from sea ice, Cardellach et al. (2008) demonstrated that these relfections do not noticeably bias RO retrievals. Cardellach and Oliveras (2016) showed that retrievals that feature reflection (from sea ice or from other sources) have lower negative refractivity biases, but these impacts are strongest in mid- and low-latitudes — the opposite of the regions shown in Fig. 6 — and do not induce a positive bias.

We investigate the source of both the positive and strong negative bias regions further in Fig. 7, which shows vertical profiles (with 200 m resolution) of the bias relative to ERA5 in three sample regions: a weak positive bias region just south of the Russian Kamchatka Peninsula, a high-stratocumulus region of the Tropics with a strong negative bias at 0.8 km, and a "control" region with the a slight negative bias (typical of GNSS RO). Comparison of the center panel of Fig. 7 to the far right (control) panel shows the dramatic impact of super-refraction on the RO refractivity profiles. The sharp divergence towards negative values from around 1.7 km to 1.2 km reflects the inability of the RO signal to capture the true refractivity in the presence of ducting. Furthermore, unlike the other two panels, the differences in refractivity across processing centers are marked. This highlights the influence of minimum height choice and filtering differences, meant to overcome super-refraction effects, on retrievals. The positive bias panel (far left), on the other hand, shows minimal difference between processing centers and no dip into negative biases at very low heights. This similarity between results from each processing center hints to a systematic bias between ERA5 and GNSS RO.

We also considered the effect of the El Niño Southern Oscillation (ENSO) on refractivity biases. The strongest El Niño covered by COSMIC-1 was the 2015 event, which peaked in strength in the autumn of that year. Comparison of the refractivity bias of UCAR relative to ERA5 for September, October, and November (SON) of both 2008 and 2015 showed no noticeable difference. We thus conclude that ENSO does not have a significant impact on the refractivity bias.

## 4 Conclusions

In this study, we have presented a characterization of differences in refractivity retrievals in the PBL from NASA JPL, UCAR, and ROM SAF. We have identified a new source of bias in the PBL: a penetration sampling bias caused by the different minimum heights assigned to an occultation by each processing center. We find that the highest (most conservative) minimum

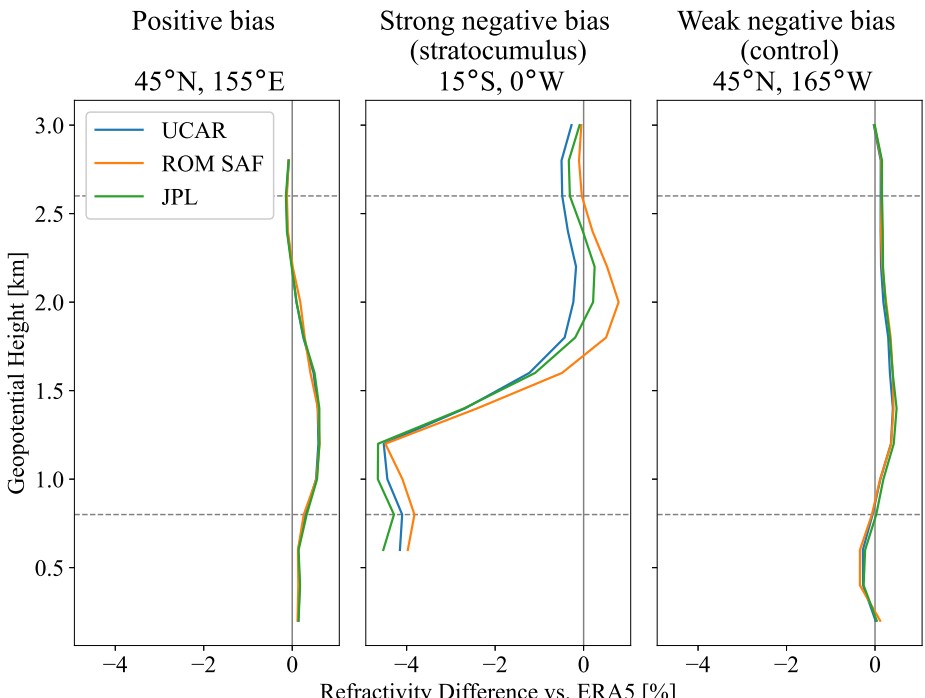

**Figure 7.** Fractional refractivity bias relative to ERA5 for all three processing centers, for three regions with distinct refractivity bias characteristics: a positive bias region (left), a strong negative bias region (center), and a "control" region with a weak negative bias (right). The thin, solid gray line indicates no bias. Sample regions were chosen using the same masking scheme as Fig. 5. Gray, dashed horizontal lines indicate the two heights of interest in this study (0.8 and 2.6 km).

height for all three processing centers is in the Tropics, with median minimum geopotential heights from 420–1000 m compared to 80–260 m for higher latitudes. We identified a direct relationship between refractivity gradient in the PBL and minimum height of retrievals: lower minimum heights correspond linearly with weaker refractivity gradients. The spatial variance of minimum penetration height and differences across processing centers illustrate the importance of considering this penetration sampling bias in the planetary boundary layer, where the retrieval minimum height is often higher than the PBL height. Within the Tropics, we find a band of lower minimum height (as low as 200-500 m, depending on the center) near the equator. This band shows a weak correlation with the ITCZ and the Pacific Cold Tongue, as explored in Appendix A.

We have investigated biases in refractivity retrievals, both across processing centers and relative to ERA5. Across processing centers, we found that UCAR had refractivity values between those of JPL and ROM SAF at 0.8 km, for the same occultations. These inter-center biases are approximately ±0.3–0.5% in the Tropics and < 0.1% outside. Sokolovskiy et al. (2010) showed that retrievals with higher minimum heights have a stronger negative bias in this region. However, we also found that UCAR had the lowest minimum heights of all three centers, for all regions of the globe. Therefore, the choice of minimum height alone

cannot be responsible for the refractivity bias between ROM SAF, JPL, and UCAR at this height. This hints at an additional source of refractivity bias in the PBL, which warrants further study.

Furthermore, we found that both JPL and ROM SAF retrieved higher refractivity than UCAR at 2.6 km, especially in the Tropics, where the inter-center bias is 0.5–1% for ROM SAF vs. UCAR and slightly smaller for JPL vs. UCAR. We hypothesize that this may indicate that the width of the radio-holographic filter of UCAR is in between that of JPL and ROM SAF, which would have greatest effect in regions of high super-refraction (i.e., the Tropics). Testing this hypothesis requires tuning retrieval parameters. In future work, we intend to vary parameters in the ROM SAF's Radio Occultation Processing Package (ROPP) (Schwärz et al., 2024) to attempt to recreate the refractivity biases identified in this work to determine their cause.

We also note that the three processing centers use different wave optics methods to overcome multi-path in the PBL: UCAR uses a phase matching technique that allows for arbitrary GNSS and LEO orbits without approximate position corrections (Jensen et al., 2004). ROM SAF and JPL use a type 2 canonical transform (Gorbunov, 2002a, b), which transforms the ray from the multivalued space of impact parameter and momentum to the single-valued space of phase and conjugate momentum, allowing the ray to be integrated without missing the ducting region (Gorbunov and Lauritsen, 2004).

Our comparison to ERA5 confirms the regions of negative refractivity bias identified by Xie et al. (2010), while demonstrating their robustness to change of model. These regions track high regions of high stratocumulus coverage with biases of up to $-4\%$ in regions with the highest cloud coverage, indicating that the negative refractivity bias is, in fact, due to super-refraction (not shown). This is confirmed in Fig. 7, where the stratocumulus region displays a very sharp refractivity bias gradient near 1.5 km, indicative of super-refraction. Biases throughout the Tropics were found to be approximately $-1.5\%$ to $+4\%$.

We also identify areas of weak (up to 0.5%) positive bias at 0.8 km. These biases seem to appear throughout from approximately 0.8 to 2.0 km in Fig. 7, and are consistent across all processing centers. Feng et al. (2020) also identified (expanded) positive bias regions at 2.0 km. The bias regions at 0.8 km appear to be strongest in the winter hemispheres, and thus are not caused by sea ice melt. All three centers have nearly identical biases relative to ERA5, including a clear positive bias "bump" that is strongest from approximately 1–1.5 km. This may be indicative of a failure of ERA5 to capture sufficiently low temperatures in these regions, as the bias is nearly identical irrespective to RO processing algorithm. We hypothesize that cold air blowing over warm water in these winter hemisphere regions may be creating model errors in these regions that bias the model towards lower refractivities. In contrast, however, all three panels of Fig. 7 show some degree of small positive bias at 2 km, agreeing with results from Feng et al. (2020). This indicates the opposite cause: that the small positive bias in these regions is due to a universal GNSS RO positive refractivity bias that is usually outweighed by the stronger negative bias. These positive bias regions warrant further investigation.

*Data availability.* The radio occultation data are available from the AWS Registry of Open Data (https://registry.opendata.aws/gnss-ro-opendata; Leroy and McVey, 2023). ERA5 profiles on model levels were selected from a horizontal grid point closest to the RO profile. These ERA5 profiles were downloaded from the Research Data Archive of the University Corporation for Atmospheric Research (https://doi.org/10.5065/ XV5R-5344). The ISCCP H-Series Cloud Properties CDR used in this study was acquired from the NOAA National Centers for Environ-

mental Information (NCEI, formerly NCDC; https://www.ncei.noaa.gov). This CDR was developed by William B. Rossow, Violeta Golea, and Cindy Pearl of The City College of New York, Alison Walker of Desert Sage Software, Inc., Alisa Young and Ken Knapp of NOAA/-NESDIS/National Centers for Environmental Information, Anand Inamdar of the Cooperative Institute for Climate and Satellites (CICS) North Carolina, and Bill Hankins of ERT Inc., Asheville NC. Development of the CDR was supported by NOAA's CDR Program. Global

Precipitation Climatology Project (GPCP) Monthly Analysis Product data was provided by the NOAA PSL, Boulder, Colorado, USA, from their website at https://psl.noaa.gov.

*Author contributions.*  SV performed the analysis and wrote all sections but the introduction. SL supervised, acquired funding, wrote the introduction, and reduced model-level RO data. CAO and SL conceptualized the project. CAO, REK, KJN, K-NW, and FX encouraged investigation into penetration sampling bias and the ITCZ and engaged in analysis discussions. All authors participated in manuscript revisions.

*Competing interests.*  The authors declare no competing interests.

*Disclaimer.*  Authors C. O. Ao, K. J. Nelson, and K.-N. Wang acknowledge that portions of this work was conducted at the Jet Propulsion Laboratory, California Institute of Technology, under a contract with the National Aeronautics and Space Administration.

*Acknowledgements.*  SV and SL were funded by NASA Decadal Survey Incubator, grant 80NSSC22K1103.

## Appendix A:  Equatorial minimum height structure

As discussed in Section 3.2, the minimum height is inversely correlated with the refractivity gradient near the PBL top. Fig. A1 shows the global distribution of minimum height for each processing center. The highest (most conservative) minimum heights appear at the outer edges of the Tropics for all processing centers, interrupted by a thin, darker blue band of lower minimum heights(deeper penetration) near the equator.

It has been shown that subsidence at the edges of the Hadley cell produces sharp refractivity gradients that result in a bias

towards higher minimum heights (poorer penetration) (Xie et al., 2012), explaining the green bands of weaker penetration along the outer edges of the Tropics. In contrast, the convection in the intertropical convergence zone (ITCZ) may reduce the vertical discontinuities in refractivity, therefore the impact of noise- and bias-inducing effects in the PBL such as atmospheric multi-path and super-refraction. To test whether this lowers the minimum height of retrievals — potentially causing the dark band of lower minimum height — we label precipitation data from NOAA's Global Precipitation and Climate Project (Adler

et al., 2003), which combines satellite, sounding, and rain gauge station data to assess global precipitation on a $2.5° \times 2.5°$ grid. The red stippling in Fig. A1 indicates bins with high (greater than $5 \text{ mm d}^{-1}$) average precipitation during the shown

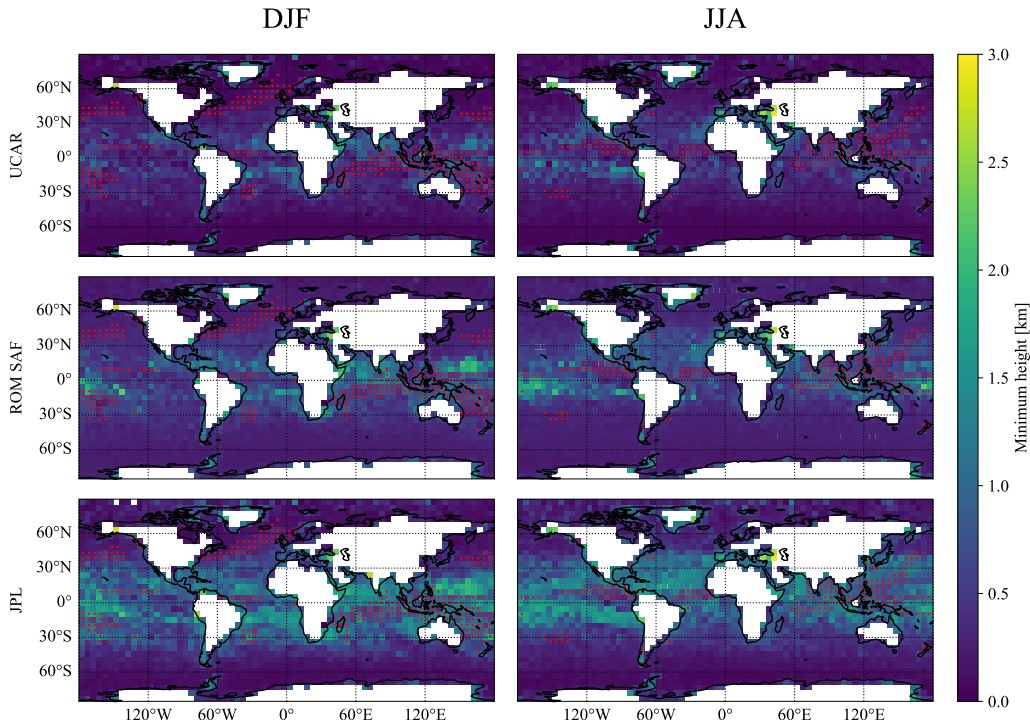

**Figure A1.** Maps of median minimum height on 5° bins for maritime occultations shared by all three processing centers for DJF (left) and JJA (right) 2008. Yellow coloring indicates more conservative (higher) minimum height. The minimum heights for all three centers are highest in the Tropics, especially for JPL, confirming the effects seen in Fig. 2. Bins with a seasonal mean precipitation of at least $5\ \mathrm{mm\ d^{-1}}$ on average are labeled with red stippling. No statistical mask is used in this figure so that fine, small-scale zonal and meridional patterns in minimum height are shown, but abnormally high or low minimum height values in individual cells — especially those along coastlines — may be due to larger noise.

seasons. The band of red stippling near the equator in each plot approximates the location of the intertropical convergence zone (ITCZ). We find that the band of lower minimum height (better penetration) only very roughly tracks the ITCZ, providing weak evidence of a possible connection between the ITCZ and the band of lower minimum height. While this band of low minimum height also roughly tracks the Pacific Cold Tongue Zhang et al. (2010); Jin (1996); Moum et al. (2013) in the eastern Pacific, this mechanism is unable to explain why the band also appears (although weaker) across the Atlantic Ocean.

## Appendix B: JPL and ROM SAF reference comparisons

Figures B1 and C1 show the biases of JPL and ROM SAF, respectively, relative to ERA5.

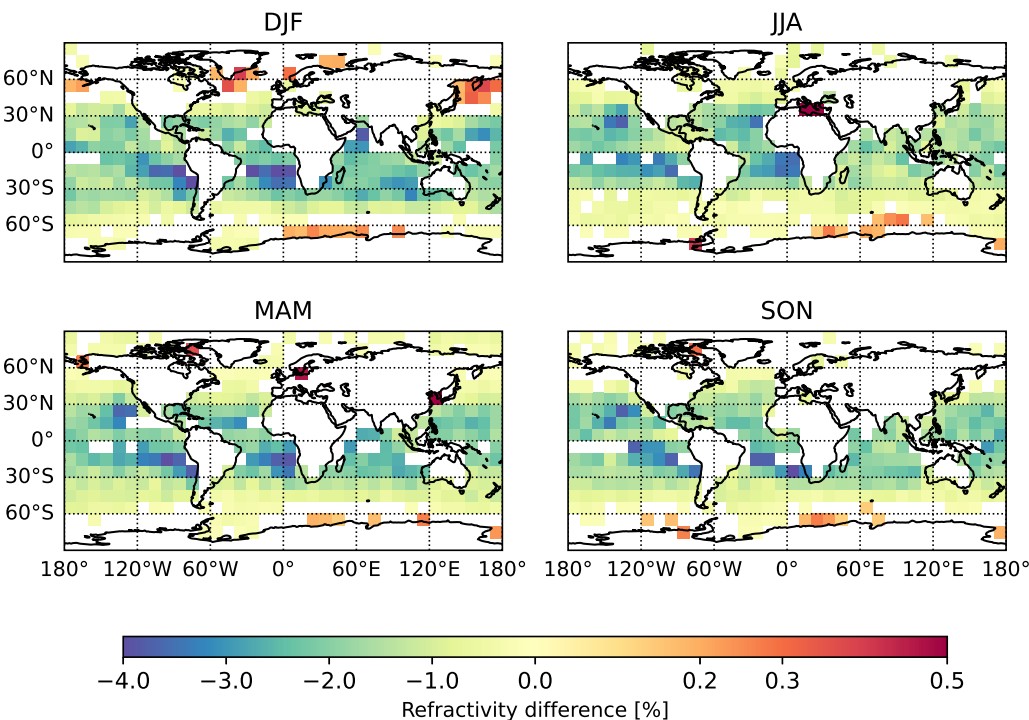

**Figure B1.** Bias between refractivity profiles retrieved from COSMIC-1 by JPL vs. those modeled by ERA5 for maritime occultations for each season of 2008, just as in Fig. 6. Differences between GNSS RO, linearly interpolated in 20 m increments. The maps have been masked to conceal bins with means below twice the standard error. The color scale is set to a two-slope norm: yellow indicates 0 bias, while the differing slopes of the negative (blue) and positive (red) color scale allow the range of the negative bias strength to be shown while also making the weaker positive bias regions visible.

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

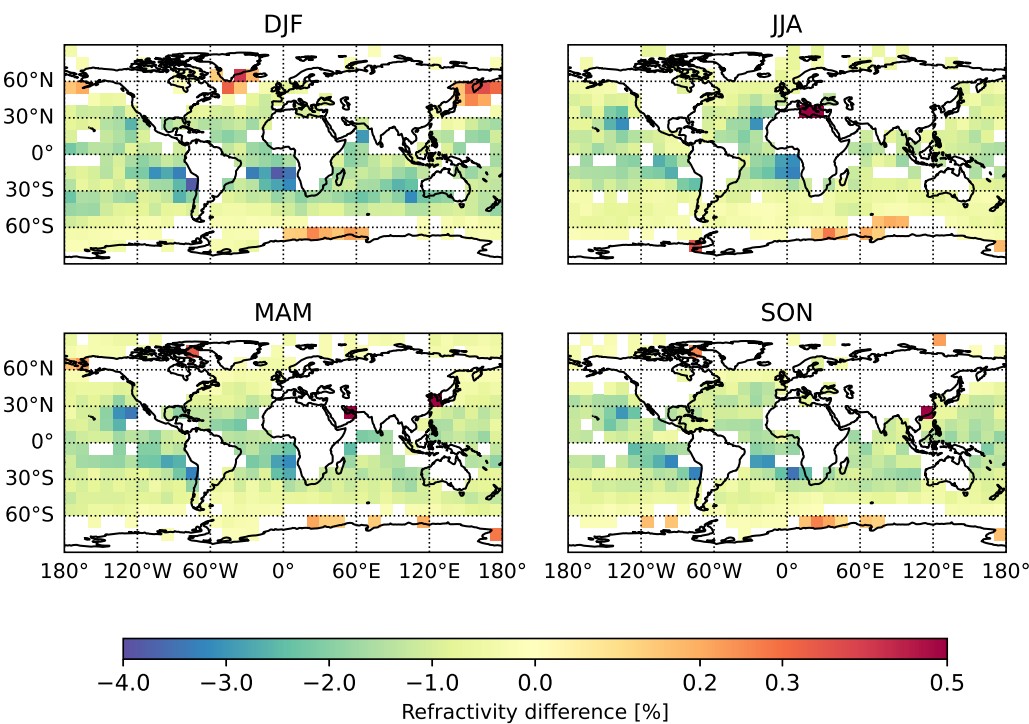

**Figure C1.** Similar to Fig. B1, comparing ROM SAF refractivity to ERA5.

Ao, C., Waliser, D., Chan, S., Li, J., Tian, B., Xie, F., and Mannucci, A.: Planetary boundary layer heights from GPS radio occultation
refractivity and humidity profiles, J. Geophys. Res., 117, D16 117, https://doi.org/10.1029/2012JD017598, 2012.

Bauer, P., Radnóti, G., Healy, S., and Cardinali, C.: GNSS Radio Occultation Constellation Observing System Experiments, Mon. Wea. Rev.,
142, 555–572, https://doi.org/10.1175/MWR-D-13-00130.1, 2014.

Born, M. and Wolf, E.: Principles of Optics, Sixth Edition, Cambridge University Press, Cambridge, England, 1980.

Cardellach, E. and Oliveras, S.: Assessment of a potential reflection flag product, Tech. rep., IEEC, Barcelona, Spain, https://rom-saf.
eumetsat.int/general-documents/rsr/rsr_23.pdf, 2016.

Cardellach, E., Oliveras, S., and Rius, A.: Applications of the Reflected Signals Found in GNSS Radio Occultation Events, https://www.
ecmwf.int/sites/default/files/elibrary/2008/7460-applications-reflected-signals-found-gnss-radio-occultation-events.pdf, 2008.

Cardinali, C. and Healy, S.: Impact of GPS radio occultation measurements in the ECMWF system using adjoint-based diagnostics, Q. J. R.
Meteorol. Soc., 140, 2315–2320, https://doi.org/10.1002/qj.2300, 2014.

Chang, H., Lee, J., Yoon, H., Morton, Y., and Saltman, A.: Performance assessment of radio occultation data from GeoOptics by comparing
with COSMIC data, Earth, Planets and Space, 74, 17, https://doi.org/10.1186/s40623-022-01667-6, 2022.

Cucurull, L.: Recent Impact of COSMIC-2 with Improved Radio Occultation Data Assimilation Algorithms, Weather and Forecasting, 38,
1829 – 1847, https://doi.org/10.1175/WAF-D-22-0186.1, 2023.

Dee, D., Uppala, S., Simmons, A., Berrisford, P., Poli, P., Kobayashi, S., Andrae, U., Balmaseda, M., Balsamo, G., Bauer, P., Bechtold, P., Beljaars, A., van de Berg, L., Bidlot, J., Bormann, N., Delsol, C., Dragani, R., Fuentes, M., Geer, A., Haimberger, L., Healy, S., Hersbach, H., Hólm, E., Isaksen, L., Kållberg, P., Köhler, M., Matricardi, M., McNally, A., Monge-Sanz, B., Morcrette, J., Park, B., Peubey, C., de Rosnay, P., Tavolato, C., Thépaut, J., and Vitart, F.: The ERA-Interim reanalysis: Configuration and performance of the data assimilation system, Q. J. R. Meteorol. Soc., 137, 553–597, 2011.

European Centre for Medium-Range Weather Forecasts: "ERA5 Reanalysis Model Level Data", https://rda.ucar.edu/datasets/dsd633006/, last accessed 1 Oct 2024, 2022.

Feng, X., Xie, F., Ao, C., and Anthes, R.: Ducting and Biases of GPS Radio Occultation Bending Angle and Refractivity in the Moist Lower Troposphere, Journal of Atmospheric and Oceanic Technology, 37, 1013 – 1025, https://doi.org/10.1175/JTECH-D-19-0206.1, 2020.

Fjeldbo, G. and Eshleman, V.: The atmosphere of Mars analyzed by integral inversion of the Mariner IV occultation data, Planetary and Space Science, 16, 1035–1059, https://doi.org/10.1016/0032-0633(68)90020-2, 1968.

Fjeldbo, G., Kliore, A., and Eshleman, V.: Neutral atmosphere of Venus as studied with Mariner-V radio occultation experiments, Astronom. J., 76, 123–140, https://doi.org/10.1086/111096, 1971.

Gleisner, H., Ringer, M., and Healy, S.: Monitoring global climate change using GNSS radio occultation, npj Climate and Atmos. Sci., 6, https://doi.org/10.1038/s41612-022-00229-7, 2022.

Golea, V., Knapp, K., Young, A., A., I., Hankins, B., and Program, N. C. D. R.: International Satellite Cloud Climatology Project Climate Data Record, H-Series ISCPP, https://doi.org/doi:10.7289/V5QZ281S, accessed on 21-08-2024, 2016.

Gorbunov, M.: Canonical transform method for processing radio occultation data in the lower troposphere, Radio Sci., 37, doi:10.1029/2000RS002 592, 2002a.

Gorbunov, M. and Lauritsen, K.: Analysis of wave fields by Fourier integral operators and their application for radio occultations, Radio Sci., 39, https://doi.org/10.1029/2003RS002971, 2004.

Gorbunov, M., Benzon, H., Jensen, A., Lohmann, M., and Nielsen, A.: Comparative analysis of radio occultation processing approaches based on Fourier integral operators, Radio Sci., 39, https://doi.org/10.1029/2003RS002916, 2004.

Gorbunov, M., Irisov, V., and Rocken, C.: Noise Floor and Signal-to-Noise Ratio of Radio Occultation Observations: A Cross-Mission Statistical Comparison, Remote Sensing, 14, https://doi.org/10.3390/rs14030691, 2022a.

Gorbunov, M., Irisov, V., and Rocken, C.: The Influence of the Signal-to-Noise Ratio upon Radio Occultation Retrievals, Remote Sensing, 14, https://doi.org/10.3390/rs14122742, 2022b.

Gorbunov, M. E.: Radio-holographic analysis of Microlab-1 radio occultation data in the lower troposphere, Journal of Geophysical Research: Atmospheres, 107, ACK 7–1–ACK 7–10, https://doi.org/10.1029/2001JD000889, 2002b.

Gorbunov, M. E.: Ionospheric correction and statistical optimization of radio occultation data, Radio Science, 37, 17–1–17–9, https://doi.org/10.1029/2000RS002370, 2002c.

Hajj, G., Kursinski, E., Romans, L., Bertiger, W., and Leroy, S.: A technical description of atmospheric sounding by GPS occultation, J. Atmos. Solar Terr. Phys., 64, 451–469, https://doi.org/10.1016/S1364-6826(01)00114-6, 2002.

Hersbach, H., Bell, B., Berrisford, P., Hirahara, S., Horányi, A., Muñoz Sabater, J., Nicolas, J., Peubey, C., Radu, R., Schepers, D., Simmons, A., Soci, C., Abdalla, S., Abellan, X., Balsamo, G., Bechtold, P., Biavati, G., Bidlot, J., Bonavita, M., De Chiara, G., Dahlgren, P., Dee, D., Diamantakis, M., Dragani, R., Flemming, J., Forbes, R., Fuentes, M., Geer, A., Haimberger, L., Healy, S., Hogan, R., Hólm, E., Janisková, M., Keeley, S., Laloyaux, P., Lopez, P., Lupu, C., Radnoti, G., de Rosnay, P., Rozum, I., Vamborg, F., Villaume, S., and Thépaut, J.: The ERA5 global reanalysis, Q. J. R. Meteorol. Soc., 146, 1999–2049, https://doi.org/10.1002/qj.3803, 2020.

Ho, S., Kirchengast, K., Leroy, S., Wickert, J., Mannucci, A., Steiner, A., Hunt, D., Schreiner, W., Sokolovskiy, S., Ao, C., Borsche, M., von Engeln, A., Foelsche, U., Heise, S., Iijima, B., Kuo, Y., Kursinski, R., Pirscher, B., Ringer, M., Rocken, C., and Schmidt, T.: Estimating the uncertainty of using GPS radio occultation data for climate monitoring: Intercomparisons of CHAMP refractivity climate records from 2002 to 2006 from different data centers, J. Geophys. Res., 114, D23 107, https://doi.org/doi:10.1029/2009JD011969, 2009.

Ho, S., Zhou, X., Kuo, Y., Hunt, D., and Wang, J.: Global Evaluation of Radiosonde Water Vapor Systematic Biases using GPS Radio Occultation from COSMIC and ECMWF Analysis, Remote Sensing, 2, 1320–1330, https://doi.org/10.3390/rs2051320, 2010.

Ho, S., Hunt, D., Steiner, A., Mannucci, A., Kirchengast, G., Gleisner, H., Heise, S., von Engeln, A., Marquardt, C., Sokolovskiy, S., Schreiner, W., Scherllin-Pirscher, B., Ao, C., Wickert, J., Syndergaard, S., Lauritsen, K., Leroy, S., Kursinski, E., Kuo, Y., Foelsche, U., Schmidt, T., and Gorbunov, M.: Reproducibility of GPS radio occultation data for climate monitoring: Profile-to-profile inter-comparison of CHAMP and climate records 2002 to 2008 from six data centers, J. Geophys. Res., 117, D18 111, https://doi.org/10.1029/2012JD017665, 2012.

Jensen, A., Lohmann, M., Benzon, H., and Nielsen, A.: Full spectrum inversion of radio occultation signals, Radio Sci., 38, https://doi.org/doi:10.1029/2002RS002763, 2003.

Jensen, A., Lohmann, M., Nielsen, A., and Benzon, H.: Geometrical optics phase matching of radio occultation signals, Radio Sci., 39, RS3009, https://doi.org/10.1029/2003RS002899, 2004.

Jin, F.-F.: Tropical ocean-atmosphere interaction, the Pacific cold tongue, and the El Niño-Southern Oscillation, Science, 274, 76–78, 1996.

Johnston, B. and Xie, F.: Characterizing Extratropical Tropopause Bimodality and its Relationship to the Occurrence of Double Tropopauses Using COSMIC GPS Radio Occultation Observations, Remote Sensing, 12, https://doi.org/10.3390/rs12071109, 2018.

Johnston, B., Xie, F., and Liu, C.: The Effects of Deep Convection on Regional Temperature Structure in the Tropical Upper Troposphere and Lower Stratosphere, J. Geophys. Res., 123, 1585–1603, https://doi.org/10.1002/2017JD027120, 2018.

Kuo, Y., Wee, T., Sokolovskiy, S., Rocken, C., Schreiner, W., Hunt, D., and Anthes, R.: Inversion and Error Estimation of GPS Radio Occultation Data, J. Meteorol. Soc. Japan, 82, 507–531, https://doi.org/10.2151/jmsj.2004.507, 2004.

Kursinski, E., Hajj, G., Hardy, K., Romans, L., and Schofield, J.: Observing tropospheric water-vapor by radio occultation using the Global Positioning System, GRL, 22, 2365–2368, https://doi.org/10.1029/95GL02127, 1995.

Kursinski, E., Hajj, G., Bertiger, W., Leroy, S., Meehan, T., Romans, L., Schofield, J., McCleese, D., Melbourne, W., Thornton, C., Yunck, T., Eyre, J., and Nagatani, R.: Initial results of radio occultation observations of Earth's atmosphere using the Global Positioning System, Science, 271, 1107–1110, https://doi.org/10.1126/science.271.5252.1107, 1996.

Kursinski, E., Hajj, G., Schofield, J., Linfield, R., and Hardy, K.: Observing Earth's atmosphere with radio occultation measurements using the Global Positioning System, J. Geophys. Res., 102, 23 429–23 465, https://doi.org/10.1029/97JD01569, 1997.

Kursinski, E., Hajj, G., Leroy, S., and Herman, B.: The GPS radio occultation technique, TAO, 11, 53–114, https://doi.org/10.3319/TAO.2000.11.1.53(COSMIC), 2000.

Lasota, E., Steiner, A., Kirchengast, G., and Biondi, R.: Tropical cyclones vertical structure from GNSS radio occultation: an archive covering the period 2001–2018, Earth Sys. Sci. Data, 12, 2679–2693, https://doi.org/10.5194/essd-12-2679-2020, 2020.

Leroy, S. and McVey, A.: GNSS Radio Occultation Data in the AWS Cloud: Utilities and Examples, https://doi.org/10.5281/zenodo.7799039, 2023.

Leroy, S., McVey, A., Leidner, S., Zhang, H., and Gleisner, H.: GNSS Radio Occultation Data in the AWS Cloud, Earth and Space Sci., 11, e2023EA003 021, https://doi.org/10.1029/2023EA003021, 2024.

Liou, Y., Pavelyev, A., Liu, S., Pavelyev, A., Yen, N., Huang, C., and Fong, C.: FORMOSAT-3/COSMIC GPS Radio Occultation Mission: Preliminary Results, IEEE Trans. Geosci. Rem. Sensing, 45, 3813–3844, https://doi.org/10.1109/TGRS.2007.903365, 2007.

Mannucci, A., Ao, C., and Williamson, W.: GNSS Radio Occultation, chap. 33, pp. 971–1013, John Wiley & Sons, Ltd, ISBN 9781119458449, https://doi.org/10.1002/9781119458449.ch33, 2020.

Moum, J., Perlin, A., Nash, J., and McPhaden, M.: Seasonal sea surface cooling in the equatorial Pacific cold tongue controlled by ocean mixing, Nature, 500, 64–67, 2013.

National Academies of Sciences, Engineering, and Medicine: Thriving on Our Changing Planet: A Decadal Strategy for Earth Observation from Space, The National Academies Press, Washington, D.C., https://doi.org/10.17226/24938, 2018.

Randel, W. and Wu, F.: Kelvin wave variability near the equatorial tropopause observed in GPS radio occultation measurements, J. Geophys. Res., 110, D03 102, https://doi.org/10.1029/2004JD005006, 2005.

Randel, W., Wu, F., and Ríos, W.: Thermal variability of the tropical tropopause region derived from GPS/MET observations, J. Geophys. Res., 108, 4024, https://doi.org/10.1029/2002JD002595, 2003.

Rocken, C., Kuo, Y., Schreiner, W., Hunt, D., Sokolovskiy, S., and McCormick, C.: COSMIC system description, Terr. Atmos. Ocean. Sci., 11, 21–52, 2000.

ROM SAF: ROM SAF Radio Occultation Climate Data Record - COSMIC, EUMETSAT SAF on Radio Occultation Meteorology, Tech. rep., https://doi.org/10.15770/EUM_SAF_GRM_0003, 2019.

Schmidt, T., Alexander, P., and de la Torre, A.: Stratospheric gravity wave momentum flux from radio occultations, J. Geophys. Res., 121, doi:10.1002/2015JD024 135, 2016.

Schreiner, W., Weiss, J., Braun, J., Chu, V., Fong, J., Hunt, D., Kuo, Y.-H., Meehan, T., Serafino, W., Sjoberg, J., Sokolovskiy, S., Ta-laat, E., Wee, T., and Zeng, Z.: COSMIC-2 Radio Occultation Constellation: First Results, Geophys. Res. Lett., 47, e2019GL086 841, https://doi.org/10.1029/2019GL086841, 2020.

Schwärz, M., Lewis, O., and Lauritsen, K.: The Radio Occultation Processing Package (ROPP) Overview, Tech. rep., Radio Occultation Meteorology Satellite Application Facility (ROM SAF), https://rom-saf.eumetsat.int/romsaf_ropp_ov.pdf, 2024.

Sievert, T., Rasch, J., Carlström, A., Mats I. Pettersson, M., and Vu, V.: Comparing reflection signatures in radio occultation measurements using the full spectrum inversion and phase matching methods, vol. 10786, p. 107860A, International Society for Optics and Photonics, SPIE, https://doi.org/10.1117/12.2325386, 2018.

Smith, E. and Weintraub, S.: The constants in the equation for atmospheric refractive index at radio frequencies, Proc. IEEE, 41, 1035–1037, 1953.

Sokolovskiy, S.: Modeling and inverting radio occultation signals in the moist troposphere, Radio Sci., 36, 441–458, https://doi.org/10.1029/1999RS002273, 2001.

Sokolovskiy, S.: Effect of superrefraction on inversions of radio occultation signals in the lower troposphere, Radio Sci., 38, 1058, https://doi.org/10.1029/2002RS002728, 2003.

Sokolovskiy, S.: Standard RO Inversions in the Neutral Atmosphere 2013 - 2020 (Processing Steps and Explanation of Data), Tech. rep., UCAR, https://www.cosmic.ucar.edu/sites/default/files/2021-09/cdaac_ro_retrieval_description_v1.pdf, 2021.

Sokolovskiy, S., Rocken, C., Hunt, D., Schreiner, W., Johnson, J., Masters, D., and Esterhuizen, S.: GPS profiling of the lower troposphere from space: Inversion and demodulation of the open-loop radio occultation signals, Geophysical Research Letters, 33, https://doi.org/10.1029/2006GL026112, 2006.

Sokolovskiy, S., Rocken, C., Schreiner, W., and Hunt, D.: On the uncertainty of radio occultation inversions in the lower troposphere, J. Geophys. Res., 115, D22 111, https://doi.org/10.1029/2010JD014058, 2010.

Sokolovskiy, S., Schreiner, W., Zeng, Z., Hunt, D., Lin, Y., and Kuo, Y.: Observation, analysis, and modeling deep radio occultation signals: Effects of tropospheric ducts and interfering signals, Radio Sci., 49, 954–970, https://doi.org/10.1002/2014RS005436, 2014.

Sokolovskiy, S., Zeng, Z., Hunt, D., Weiss, J., Braun, J., Schreiner, W., Anthes, R., Kuo, Y., Zhang, H., Lenschow, D., and Vanhove, T.: Detection of superrefraction at the Top of the Atmospheric Boundary Layer from COSMIC-2 Radio Occultations, JAOT, 41, 65–78, https://doi.org/10.1175/JTECH-D-22-0100.1, 2024.

Steiner, A., Hunt, D., Ho, S., Kirchengast, G., Mannucci, A., Scherllin-Pirscher, B., Gleisner, H., von Engeln, A., Schmidt, T., Ao, C., Leroy, S., Kursinski, E., Foelsche, U., Gorbunov, M., Heise, S., Kuo, Y., Lauritsen, K., Marquardt, C., Rocken, C., Schreiner, W., Sokolovskiy,

S., Syndergaard, S., and Wickert, J.: Quantification of structural uncertainty in climate data records from GPS radio occultation, ACP, 13, 1469–1484, https://doi.org/10.5194/acp-13-1469-2013, 2013.

Steiner, A., Ladstädter, F., Randel, W., Maycock, A., Fu, Q., Claud, C., Gleisner, H., Haimberger, L., Ho, S., Keckhut, P., Leblack, T., Mears, C., Polvani, L., Santer, B., Schmidt, T., Sofieva, V., Wing, R., and Zou, C.: Observed Temperature Changes in the Troposphere and Stratosphere from 1979 to 2018, JCLI, 33, 8165–8194, https://doi.org/10.1175/JCLI-D-19-0998.1, 2020.

Syndergaard, S., Nielsen, J., and Lauritsen, K.: Algorithm Theoretical Baseline Document: Level 1B bending angles, Tech. rep., Radio Occultation Meteorology Satellite Application Facility (ROM SAF), Lyngbyvej 100, Copenhagen, Denmark, https://rom-saf.eumetsat.int/product_documents/romsaf_atbd_ba.pdf, 2020.

Syndergaard, S., Nielsen, J., and Lauritsen, K.: Algorithm Theoretical Baseline Document: Level 2A refractivity profiles, Tech. rep., Radio Occultation Meteorology Satellite Application Facility (ROM SAF), Lyngbyvej 100, Copenhagen, Denmark, https://rom-saf.eumetsat.int/

product_documents/romsaf_atbd_ref.pdf, 2021.

Teixeira, J., Piepmeier, J., Nehrir, A., Ao, C., Chen, S., Clayson, C., Fridlind, A., Lebsock, M., McCarty, W., Salmun, H., Santanello, J., Turner, D., Wang, Z., and Zeng, X.: Toward a Global Planetary Boundary Layer Observing System: The NASA PBL Incubation Study Team Report, Tech. rep., National Aeronautics and Space Administration, 2021.

Tsuda, T., Nishida, M., Rocken, C., and Ware, R.: A Global Morphology of Gravity Wave Activity in the Stratosphere Revealed by the GPS

Occultation Data (GPS/MET), J. Geophys. Res., 105, 7257–7273, 2000.

UCAR: FORMOSAT-3/COSMIC-1 2021 Reprocessing Data Release, https://data.cosmic.ucar.edu/gnss-ro/cosmic1/repro2021/UCAR_COSMIC1_2021_Repro_Notes.pdf, accessed on 20-04-2025, 2022.

Vergados, P., Mannucci, A., and Su, H.: A validation study for GPS radio occultation data with moist thermodynamic structure of tropical cyclones, J. Geophys. Res., 118, 9401–9413, https://doi.org/10.1002/jgrd.50698, 2013.

Vergados, P., Ao, C., Mannucci, A., and Kursinski, E.: Quantifying the Tropical Upper Tropospheric Warming Amplification Using Radio Occultation Measurements, Earth and Space Sci., 8, e2020EA001 597, https://doi.org/10.1029/2020EA001597, 2021.

Verkhoglyadova, O., Leroy, S., and Ao, C.: Estimation of Winds from GPS Radio Occultations, J. Atmos. Ocean. Tech., 31, 2451–2461, https://doi.org/10.1175/JTECH-D-14-00061.1, 2014.

Vorob'ev, V. and Krasil'nikova, T.: Estimation of the accuracy of the atmospheric refractive index recovery from Doppler shift measurements

at frequencies used in the NAVSTAR system, Phys. Atmos. Ocean., 29, 602–609, 1994.

Wang, K., de la Torre Juárez, M., Ao, C., and Xie, F.: Correcting negatively biased refractivity below ducts in GNSS radio occultation: an optimal estimation approach towards improving planetary boundary layer (PBL) characterization, Atmos. Meas. Tech., 10, 4761–4776, https://doi.org/10.5194/amt-10-4761-2017, 2017.

Wang, K., Ao, C., and de la Torre Juárez, M.: GNSS-RO Refractivity Bias Correction Under Ducting Layer Using Surface-Reflection Signal, Remote Sens., 12, 359, https://doi.org/10.3390/rs12030359, 2020.

Wang, K., Ao, C., Morris, M., Hajj, G., Kurowski, M., Turk, F., and Moore, A.: Joint 1DVar retrievals of tropospheric temperature and water vapor from Global Navigation Satellite System radio occultation (GNSS-RO) and microwave radiometer observations, Atmos. Meas. Tech., 17, 583–599, https://doi.org/10.5194/amt-17-583-2024, 2024.

Wang, L. and Alexander, M.: Global estimates of gravity wave parameters from GPS radio occultation temperature data, J. Geophys. Res., 115, D21 122, https://doi.org/doi:10.1029/2010JD013860, 2010.

Xie, F.: An Approach for Retrieving Marine Boundary Layer Refractivity from GPS Occultation Data in the Presence of Superrefraction, JAOT, 23, 1629–1644, https://doi.org/10.1175/JTECH1996.1, 2006.

Xie, F., Wu, D., Ao, C., Kursinski, E., Mannucci, A., and Syndergaard, S.: Super-refraction effects on GPS radio occultation refractivity in marine boundary layers, Geophys. Res. Lett., 37, L11 805, https://doi.org/10.1029/2010GL043299, 2010.

Xie, F., Wu, D., Ao, C., Mannucci, A., and Kursinski, E.: Advances and limitations of atmospheric boundary layer observations with GPS occultation over the southeast Pacific Ocean, Atmos. Chem. Phys., 12, 903–918, https://doi.org/10.5194/acp-12-903-2012, 2012.

Zeng, Z., Sokolovskiy, S., Schreiner, W., Hunt, D., Lin, J., and Kuo, Y.-H.: Ionospheric correction of GPS radio occultation data in the troposphere, Atmospheric Measurement Techniques, 9, 335–346, https://doi.org/10.5194/amt-9-335-2016, 2016.

Zeng, Z., Sokolovskiy, S., Hunt, D., Weiss, J., Braun, J., Schreiner, W., Anthes, R., Kuo, Y., Zhang, H., Lenschow, D., and Vanhove, T.: Estimation of the heights of superrefraction layers from radio occultation signals, https://www.cosmic.ucar.edu/sites/default/files/2024-09/13%20-%20Zeng_Zhen_2024.09.16.pptx.pdf, 2024.

Zhang, W., Li, J., and Zhao, X.: Sea surface temperature cooling mode in the Pacific cold tongue, Journal of Geophysical Research: Oceans, 115, https://doi.org/10.1029/2010JC006501, 2010.