# Peer review of "The Impact of Differences in Retrieval Algorithms between Processing Centers on GNSS Radio Occultation Refractivity Retrievals in the Planetary Boundary Layer"

_EGUsphere, 2024_

## Author Response (AR1)

Thank you for your consideration of our paper revisions. We have responded to each of the referee comments and revised our paper accordingly. These comments are listed below.

**Referee #1:**

We thank the anonymous referee for their thorough and insightful feedback. We respond to each individual comment below.

We would like to clarify that our paper in meant to characterize retrieval differences between processing centers in the PBL, not to identify the causes of these differences. While similar studies have investigated retrieval differences in the upper troposphere/lower stratosphere, this work is, to our knowledge, the first to characterize these differences in the PBL. We agree that the retrieval differences across processing centers are due to differences in processing center algorithms and quality control. However, in order to identify the causes of retrieval differences, we must first identify what the differences are. This is the purpose of this paper. We have added language throughout the paper, especially in the introduction and conclusion, to clarify this.

L.8. ...and UCAR (420 km).

-- 420 m.

We have corrected this typo.

L. 36. Downward vertical gradients in the microwave index of refraction can become so strong in a layer of the atmosphere that rays are ducted, rendering the ducting layer invisible to external rays, such as those from the GNSS transmitter: a phenomenon known as super-refraction. RO can measure the atmosphere above and below such ducts but never inside them.

-- The inverse problem in the presence of ducts has no unique solution, and this affects also the profile below the duct. I suggest putting this in stricter terms, along the lines of the references:

1. Sokolovskiy, S. V. (2003), 'Effect of super refraction on inversions of radio occultation

signals in the lower troposphere', Radio Sci. 38(3), 1058. DOI: 10.1029/2002RS002728

2. Sokolovskiy, S., Schreiner, W., Zeng, Z., Hunt, D., Lin, Y.-C. and Kuo, Y.-H. (2014), 'Observation, analysis, and modeling of deep radio occultation signals: Effects of tropospheric ducts and interfering signals', Radio Sci. 49(10), 954–970. DOI: 10.1002/2014RS005436

3. Sokolovskiy, S., Zeng, Z., Hunt, D. C., Weiss, J.-P., Braun, J. J., Schreiner, W. S., Anthes, R. A., Kuo, Y.-H., Zhang, H., Lenschow, D. H. and Vanhove, T. (2024), 'Detection of Superrefraction at the Top of the Atmospheric Boundary Layer from COSMIC-2 Radio Occultations', J. Atmos. Ocean. Technol. 41(1), 65–78. DOI: 10.1175/jtech-d-22-0100.1

4. Xie, F., Syndergaard, S., Kursinski, E. R. and Herman, B. (2006), 'An Approach for Retrieving Marine Boundary Layer Refractivity from GPS Occultation Data in the Presence of Superrefraction', J. Atmos. Ocean. Technol. 23(12), 1629–1644.

We thank the referee for this feedback. We have updated this description of super-refraction to be more precise, along the lines of the suggested papers. This section now reads as follows:

*The problem of super-refraction is based in the physics of the RO observation. Vertical gradients in the microwave index of refraction can become so sharp that rays with tangent points within the ducting layer have a radius of curvature smaller than the radius of the Earth. This causes total internal reflection inside the ducting layer, rendering the ducting layer invisible to external rays: a phenomenon known as super-refraction. (Sokolovskiy, 2003; Sokolovskiy et al., 2014, 2024) There is no single solution to retrieve a refractivity profile with super-refraction present from a bending angle profile (Xie, 2006). While RO performance remains unaffected above the ducting layers, the "invisibility" of the duct results in large negative biases in retrieved refractivity and even larger fractional biases in retrieved water vapor below the duct (Xie, 2006).*

L. 62. The generation 3 missions, including the six satellites of COSMIC-2 and the satellites of the commercial RO provider PlanetiQ, obtain median SNRs of roughly 2000 V/V (1 Hz). These new, exceptionally large SNRs allow RO signal tracking deeper into the PBL than ever before, even in the presence of extreme bending and super-refraction.

-- See the discussion of the SNR:

5. Gorbunov M., Irisov V., and Rocken C. Noise Floor and Signal-to-Noise Ratio of Radio

Occultation Observations: A Cross-Mission Statistical Comparison. Remote Sensing. 2022, 14(3), 691, DOI: 10.3390/rs14030691

6. Gorbunov M., Irisov V., and Rocken C. The Influence of the Signal-to-Noise Ratio upon Radio Occultation Retrievals, Remote Sensing. 2022, 14(12), 2742; DOI: 10.3390/rs14122742.

We thank the referee for this feedback. However, we note that information about the SNR of generation 3 missions was included in the introduction only for context. We have clarified this section along the lines of the suggested references to read:

*The generation 3 missions, including the six satellites of COSMIC-2 and the satellites of the commercial RO provider PlanetiQ, obtain median SNRs of roughly 1500 V/V (1 Hz) (Schreiner et al. 2020). (Gorbunov et al. 2022) introduced two forms of normalized SNR that to assess the noise floor of each individual profile, showing that a higher SNR leads to better penetration in the PBL. Gorbunov et al. 2022a confirmed that COSMIC-2 has a better noise floor than older missions. These new, exceptionally large SNRs allow deeper signal tracking that ever before, even in the presence of extreme bending and super-refraction, reducing refractivity biases.*

L.65. Very recent results show that these high-SNR RO soundings enable the detection of the presence of super-refraction (Sokolovskiy et al., 2024) and the critical refractional radius1 that defines the super-refraction duct, so long as the layers are not attached to Earth's surface (Zeng et al., 2024).

-- What exactly does it mean: "not attached to Earth's surface"? Do you want to say that we only know the refractive radius rather than the geometric height of the duct?

"Not attached to the surface" is meant to distinguish between surface ducts (where the duct lies along the surface of the Earth) and elevated ducts (where the bottom of the duct is elevated above the Earth). We have changed "so long as the layers are not attached to Earth's surface" to read "*so long as the bottom of the ducting layer is elevated above Earth's surface.*"

L. 68. UCAR has recently begun publishing level 1 (calibrated excess phase) COSMIC-2 RO data with a super-refraction detection flag and a value for the duct refractional

radius; those data can then be used in a retrieval of water vapor from RO data that seeks to be unbiased.

-- Any references regarding the use of these data for water vapor retrieval?

This data has not yet been used in water vapor retrievals, but was suggested by Xie et al. 2006 [1]. We have edited the text in lines 43-45 to explain this. We have clarified this text to read

*A general approach to mitigating the negative biases below super-refraction ducts using additional external information to correct the super-refraction impact height was suggested by (Xie et al. 2006) Examples of external information in proposed unbiased retrievals include a measurement of total column water vapor (Wang et al. 2017), the synchronously received signal reflected off the ocean surface (Wang et al. 2020), and collocated nadir radiance microwave data (Wang et al. 2024).*

[1] Xie, F.: An Approach for Retrieving Marine Boundary Layer Refractivity from GPS Occultation Data in the Presence of Superrefraction, JAOT, 23, 1629–1644, https://doi.org/10.1175/JTECH1996.1, 2006

L. 71. While unbiased retrievals of water vapor in the PBL from RO data have yet to be published, the existing retrieval algorithms can be examined for systematic and structural refractivity errors in the PBL.

-- There are no unbiased retrieval algorithms? Or they are under development, but not yet published? Provide here references on the existing water vapor retrieval algorithms. Use more precise language.

See above for comment on unbiased water vapor retrievals. We have changed the language to reflect that we are looking to examine the biased retrievals. The possibility of unbiased retrievals in the future is added only for context in the introduction and does not impact the results of this work. This work is meant to consider structural differences in retrievals as a result of existing retrieval algorithms, not algorithms that may exist in the future.

L. 72. The components of existing retrieval systems that can induce bias include implementations of navigation message demodulation, radio-holographic filtering and the smoothing it begets, the wave optics retrieval itself, and the approach to cutting off an RO signal low in the atmosphere when the signal

becomes too weak to be of use.

*-- Provide here the references on the biases from each item: 1) radio-holographic filtering; 2) the wave optics retrieval; 3) cutting off.*

We have added citations for

1. Removing navigation bits:
   Sokolovskiy, S., Rocken, C., Hunt, D., Schreiner, W., Johnson, J., Masters, D., and Esterhuizen, S.: GPS profiling of the lower troposphere from space: Inversion and demodulation of the open-loop radio occultation signals, Geophysical Research Letters, 33, https://doi.org/10.1029/2006GL026112, 2006.

2. Radioholographic filtering:
   Sokolovskiy, S., Rocken, C., Schreiner, W., and Hunt, D.: On the uncertainty of radio occultation inversions in the lower troposphere, J. Geophys. Res., 115, D22 111, https://doi.org/10.1029/2010JD014058, 2010.

3. Wave optics retrieval: Radioholographic filtering is a necessary part of wave optics retrievals; RF filtering produces the bias in wave optics retrievals. We have edited the text to reflect that these are really one source of bias; it now reads
   *"radio-holographic filtering in wave optics retrievals and the smoothing it begets (Sokolovskiy et al. 2010)"*

4. Cutting off
   This is also explained in
   Sokolovskiy, S., Rocken, C., Schreiner, W., and Hunt, D.: On the uncertainty of radio occultation inversions in the lower troposphere, J. Geophys. Res., 115, D22 111, https://doi.org/10.1029/2010JD014058, 2010.

L. 91. Their [FSI, CT2, PM] performances are similar but not identical.

*-- Provide references. My experience tells that the bias is mostly defined by the aforementioned implementation options rather than by the choice of a specific FIO-based algorithm.*

This is commonly known in the RO community. The paper which introduced CT2 [1] does compare the performance of geometric optics, back propagation + a canonical transform, and CT2, finding that they achieve similar results. Furthermore, [2] compared FSI and PM, finding similar results. We have added both these citation to the end of this sentence. However, to our knowledge, no work has been published intercomparing FSI, CT2, and PM.

[1] Gorbunov, M. and Lauritsen, K.: Analysis of wave fields by Fourier integral operators and their application for radio occultations, Radio Sci., 39, https://doi.org/10.1029/2003RS002971, 2004.

[2] Sievert, T., Rasch, J., Carlström, A., Mats I. Pettersson, M., and Vu, V.: Comparing reflection signatures in radio occultation measurements using the full spectrum inversion and phase matching methods, vol. 10786, p. 107860A, International Society for Optics and Photonics, SPIE, https://doi.org/10.1117/12.2325386, 2018.

L. 160. While, in principle, Eq. 1 is integrated up to an impact parameter of infinity, in reality each processing center must make a choice of what maximum altitude to integrate up to and handle contributions due to layers higher in the atmosphere with great care. Typically, this is somewhere in the thermosphere.

-- The authors write once again the Abel transform pair, which can be found in hundreds of publications. The whole paper does not say a word about ionospheric correction and statistical optimization! These algorithms, depending on many parameters and options, constitute one of the most important causes of cross-center differences!

The Abel integral is included in this paper as a critical piece of contextual information to explore sources of bias and error in retrievals. However, we have shortened this section of RO introduction which has been published elsewhere, including removing the forward Abel integral. We consider ionospheric corrections to be a secondary effect, as we are interested only in the planetary boundary layer in this work. We have added context about ionospheric corrections to this section which reads,

*Differences in ionospheric corrections across processing centers may also introduce biases. All three of the processing centers use a linear combination of L1 and L2 channel bending angles to perform their ionospheric corrections (Vorobev et al. 1994) (ROM SAF uses an optimized combination from (Gorbunov 2002b) ), while extrapolating the correction terms downward lower in the atmosphere. JPL makes the transition between linear combination and extrapolation at 10 km, while UCAR and ROM SAF set the transition to extrapolation at a quality control height defined dynamically on L2 data. (Ho et al. 2012). However, we anticipate the impact of differences in ionospheric correction to be a secondary impact in the PBL: Zeng et al. 2016 found that varying the height of this transition between 0-25km caused a refractivity bias of, at most, approximately 5% in extreme cases (e.g. extrapolation transition heights below 10km), and, on average, roughly a tenth of a percent. The average biases due to ionospheric correction are therefore considerably smaller than the retrieval biases identified in this work.*

L. 175. UCAR uses a phase matching technique (Kuo et al., 2004; Sokolovskiy, 2001, 2003; Sokolovskiy et al., 2010), while JPL (Hajj et al., 2002) and ROM SAF use a type-2 canonical transform (Schwärz et al., 2024; Syndergaard et al., 2020, 2021). These techniques produce similar but not identical results, and will produce refractivity retrieval differences systematic to each processing center.

-- The differences are not so much due to different FIO algorithms, but due to the other details. See above.

Correct the URLs of the references.

Syndergaard et al., 2020:

https://rom-saf.eumetsat.int/product_documents/romsaf_atbd_ba.pdf

Syndergaard et al., 2021:

https://rom-saf.eumetsat.int/product_documents/romsaf_atbd_ref.pdf

We thank the anonymous referee and have corrected these URLs. We agree that refractivity biases due to differences in FIO algorithm are very small. We note that the similarity of results from PM, FSI, and CT2 are explained in the introduction. Therefore, we have decided to remove this sentence entirely. This section now reads

*"In these scenarios, radio-holographic filtering techniques are applied in order to reduce the influence of noise in a non-linear retrieval algorithm. UCAR uses a phase matching technique (Kuo et al. 2004, Sokolovskiy 2001, Sokolovskiy 2003, Sokolovskiy et al. 2010}, while JPL (Hajj et al. 2002) and ROM SAF use a type-2 canonical transform (ROMSAF 2024a, ROM SAF 2020a, ROM SAF 2021a). As explained in Section 1, (Sokolovskiy et al. 2010) hypothesized that wider radio-holographic filter widths result in a larger (more positive) refractivity retrieval."*

L. 178. This effect is especially strong around a geopotential height of 2–3 km.

-- Once you speak about a roughly approximate height, it is totally unnecessary to distinguish between the geometrical and geopotential height, which have much smaller differences especially at 2-3 km. Note, the geopotential height is measured in gpkm.

We agree that there is no need for a distinction. We have removed the "geopotential" to read

*This effect is especially strong around a height of 2-3 km.*

Note we have kept the "geopotential" in the next sentence, as our calculation of height is the definition of geopotential (not geometrical) height.

*We have calculated geopotential height by dividing the geopotential values in the refractivityRetrieval files by the WMO standard value for gravity, $g_0$ = 9.80665 J $^{-1}$/m$^{-1}$.*

L. 185. Furthermore, quality control and choice of minimum height create retrieval differences between processing centers. Quality control measures discard occultations at any of the three processing steps that produce sufficiently anomalous occultations. JPL, for example, discards refractivity retrievals with a refractivity difference between RO and that predicted by a reanalysis model greater than 10%. What constitutes "anomalous" is determined by each processing center and would be expected to produce additional systematic sampling biases.

-- The choice of the minimum height is often performed dynamically based on the amplitude after the FIO processing. Mention this, too. Write more about QC and criteria of events, "sufficiently anomalous" to be rejected. Are there any studies of the sampling differences that "would be expected"?

We have added the following information about quality control between the three centers.

*ROM SAF out retrievals with impact parameter noise above a certain threshold, as well as rejecting retrievals with bending angle or refractivity values sufficiently divergent from ionospheric correction and climate models. UCAR similarly screens out retrievals with bending angle or refractivity values that sufficiently diverge from climate models, as those with a high standard deviation of relative difference in bending angle or refractivity with the climate model. UCAR also rejects occultations with sufficiently high differences in bending angle between the L1 and L2 channels. JPL quality control rejects occultations with negative bending angles, as well as high-altitude bending angles sufficiently divergent from a fit that is exponential with height. Similar to ROM SAF and UCAR, JPL also rejects retrievals with refractivity values sufficiently divergent from a climatology model at any height. JPL is distinct from the other two centers in that it also screens retrievals with temperatures sufficiently different from the climate model. For more a more detailed comparison of quality control between the three processing centers, see Ho et al. 2012 and Sokolovskiy 2020.*

Ho et al. 2012 presented results showing the mean number of "profiles" (retrievals of bending angle, refractivity, dry temperature, dry pressure, and dry geopotential height) by latitude bin for different processing centers for CHAMP 2007 occultations. In their analysis, UCAR seems, in general, to have slightly more retrievals pass quality control than ROM SAF, while the relative

number of JPL retrievals relative to UCAR and ROM SAF varies significantly with latitude. Note this work uses an older reprocessing than the UCAR retrievals used in our work. Similarly, in our work, UCAR has a higher number of refractivity retrievals which pass quality control. While their work does not quantify the total overlap in retrievals between the three centers, it is clear that the vast majority of retrievals that pass quality control for one center also pass for the other two, which is also true of our work. To our knowledge, there is no study looking at the expected differences in number of retrievals due to differences in quality control across the three processing centers. We have added a sentence summarizing this to the end of the "Quality control: Overlap of Retrievals by Center" section, which reads

*The vast majority of retrievals that pass quality control for one processing center also pass for at least one of the other two centers, confirming a similar investigation by Ho et al. 2012.*

L. 200. This creates a small correlation between RO retrievals, especially from UCAR, and the reanalysis.

-- Any qualitative estimates of the "small" correlation?

We are unsure whether the referee really means "qualitative" or "quantitative". To our knowledge, there is no published quantitative estimate of the correlation between RO retrievals and ERA5 (or other reanalysis or NWP). Cucurull 2023 describes how COSMIC-2 retrievals (to our knowledge, similar work has not been published for COSMIC-1) are assimilated. This work describes quantitively, for example, that 90% of RO observations are not assimilated in the PBL. However, the work does not compute a correlation between RO and NWP or reanalyses, and, rather, illustrates that the correlation between RO and reanalyses in the PBL is small, as reanalyses do not assimilate RO below super-refractive layers (where it is known to be unreliable). We have edited the text to read as follows:

*This creates a small correlation between RO retrievals, especially from UCAR, and the reanalysis. We anticipate this correlation having little impact on our results, as quality control in data assimilation typically prevents radio occultation data from being assimilated below super-refractive layers (Cucurull 2023).*

Cucurull, L.: Recent Impact of COSMIC-2 with Improved Radio Occultation Data Assimilation Algorithms, Weather and Forecasting, 38, 1829 – 1847, https://doi.org/10.1175/WAF-D-22-0186.1, 2023.

L. 211. … only of only …

We have corrected this typo.

L. 216. Differences in quality control between different processing centers at different stages—from the calibrated phase vs. time (level 1a data), to bending angle vs. impact parameter (level lb), to refractivity retrieval vs. geopotential height (level 2)—each step in processing includes checks to discard anomalous occultations. These checks may include, among other strategies, comparison of retrievals with models or using noise thresholds.

-- Any references describing different QC procedures?

See comment for L. 185.

L. 221. The sage green center region in the center of the figure shows that 67.2% of the occultations were processed by all three centers. This indicates a strong level of agreement in quality control.

-- You have not formulated any criteria for "strong" of "weak" level of agreement. Therefore, the second sentence is unnecessary. Just say it is 67.2%.

We agree and have removed this second sentence.

L. 223. The higher processing rate of UCAR may be a result of COSMIC-1 being a UCAR-affiliated mission. ROM SAF and UCAR share calibrated phase data; which may explain why they share a higher percentage of retrievals not processed by JPL (15.4%). This indicates a difference in quality control at between UCAR and JPL at the calibrated phase level.

-- How do we know that there is a QC difference at the calibrated phase level? Obviously, a stricter QC at any processing stage will result in a higher rejection rate. Remove "at" in "in quality control at between".

We do not know for sure whether the difference in number of refractivity retrievals comes from QC differences at the calibrated phase stage, bending angle stage, refractivity retrieval, or (more likely), a combination of the three. We do notice that UCAR and ROM SAF share a higher percentage of retrievals not processed by JPL than either JPL and UCAR or JPL and ROM SAF share. This indicates a similarity in QC between UCAR and ROM SAF that is not shared by JPL in at least one of the

three processing stages. Since ROM SAF and UCAR share calibrated phase files, their QC at the calibrated phase stage is identical. Therefore, we hypothesize that the high overlap in retrievals from UCAR and ROM SAF may indicate a difference in QC at the calibrated phase stage by JPL. To emphasize that this is a potential cause, not a known one, we have edited this sentence to read "This may indicate a difference in quality control between UCAR and JPL at the calibrated phase level." We have also removed the "at" in "quality control at between."

L. 229. The minimum height allowed by retrievals is a critical characteristic of quality control in RO processing. As a ray penetrates  deep into the atmosphere, effects such as super-refraction layers, atmospheric multi-path, topography, and code demodulation all cause a decrease in SNR. Retrieval algorithms must therefore make a choice to truncate the occultation at some minimum altitude.

-- Any references describing different cut-off procedures? E.g. cut-off from straight-line tangent altitude, cut-off from CT amplitude etc. Do not replace specific analysis by obvious statements like this: "Each processing center makes different choices about the parameters determining how deep an occultation is allowed to penetrate". Note, there is a trade-off between the penetration depth and retrieval accuracy. It would be interesting to complements Figure 2 with a comparison of retrieval with, e.g. ECMWF re-analyses.

We have added information from each of the three processing centers about their choice of minimum height, with references for further information. This reads as follows:

*JPL minimum heights are set by fitting a step function to the canonical transform amplitude, truncating at the boundary of a large step in amplitude (Ao et al. 2012). UCAR determines minimum height separately for open and closed loop data: closed loop minimum heights are determined using a threshold for the difference in the filtered L1 Doppler shift and that of a model; closed loop minimum heights are determined from SNR (Sokolovskiy 2020, Sokolovskiy et al. 2010). ROM SAF truncates when the L1 amplitude is too weak, or when the smoothed bending angle near the surface is larger 0.1 rad (ROM SAF 2020a).*

We agree that higher penetration depth can lead to lower retrieval accuracy. We are not clear what the referee is referring to, as Fig. 2 shows penetration depths separated by processing center, but ECMWF reanalyses do not have a penetration depth. We do, however, compare UCAR refractivity to an ECMWF reanalysis in Fig 7. The relationship between minimum height and refractivity bias is shown by comparing Fig. 4 to Fig. 7.

L. 271. The band of red stippling near the equator in each plot approximates the location of the intertropical convergence zone (ITCZ). We find that the band of lower minimum height (better penetration) roughly tracks the ITCZ, while the highest minimum heights are in the subsiding branches of the Hadley cells.We hypothesize that subsidence produces sharp refractivity gradients that result in a bias towards higher minimum heights (poorer penetration), while convection along the 275 ITCZ reduces the vertical discontinuities in refractivity, causing quality control measures to allow processing of occultations deeper into the PBL.

-- I don't see much correlation between red stipplings and penetration. The above statements are not confirmed by any qualitative analysis nor references. Either present more quantitative analysis confirming your statement, or remove the red stipplings.

After further review, we agree that the connection between the ITCZ (the red stipplings near the equator) and the penetration depth is not strong. We have moved this to an appendix and discussed the ITCZ, as well as the Pacific Cold Tongue, as features being very loosely correlated to this band of deeper penetration.

L. 301. The vertical profile of the rightmost panel also shows that the positive bias of ROM SAF is very large in the Tropics at 2.6 km and shrinking quickly down to 0.8 km, so the positive bias of ROM SAF in the Tropics may really be more indicative of the strength of the positive bias at higher geopotential heights.

-- I don't understand the last part of this statement. Why should the bias at 2.6 km be indicative of bias at larger heights? Why not simply plot the bias at larger heights?

We were referring to the positive bias at 0.8 km being indicative of stronger bias at larger heights (e.g. the strong bias at 2.6 km). We have clarified this to read

*Similarly, the vertical profile of the rightmost panel shows that the positive bias of ROM SAF is very large in the Tropics at 2.6 km, small but positive at 0.8 km, and near zero outside the Tropics at both heights. We note that, given the strength of the ROM SAF positive bias at 2.6 km in the Tropics, the positive bias at 0.8 km may be more indicative of strength of the positive bias at higher geopotential heights than of true retrieval differences at that height.*

**Referee #2:**

We thank the anonymous referee for their thoughtful and detailed feedback.

We would like to clarify that our paper in meant to characterize retrieval differences between processing centers in the PBL, not to identify the causes of these differences. While similar studies have investigated retrieval differences in the upper troposphere/lower stratosphere, this work is, to our knowledge, the first to characterize these differences in the PBL. We agree that the retrieval differences across processing centers are due to differences in processing center algorithms and quality control. However, in order to identify the causes of retrieval differences, we must first identify what the differences are. This is the purpose of this paper. We have added language throughout the paper, especially in the introduction and conclusion, to clarify this.

We are not clear what the referee means by comparison with a reference. We do compare the RO retrievals to an ERA5 reanalysis. Other data sources cannot provide sufficient time or global coverage to compare to RO in this analysis. Radiosondes and radar profiling stations, for example, do not have sufficient coverage over oceans. Other satellite remote sensing techniques lack the vertical resolution to compare to RO in the PBL. While Fig. 7 shows the comparison between UCAR and ERA5, we have added supplementary figures showing similar plots for JPL vs. ERA5 and ROM SAF vs. ERA5. These are shown in the second Appendix.

We respond to in-line comments below.

- P1L8: 420m I assume

We thank the referee; we have corrected this typo.

- P2L49: The local spherical symmetry limitation has been partly corrected, at least at ECMWF, by ray tracing through the lowest atmospheric layers.

We believe the spherical symmetry the referee is referring to is larger scale than the spherical symmetry assumption in required for accurate retrievals in the PBL. We have clarified this sentence to reflect that that were are interested particularly in spherical symmetry in RO inversions (as this impacts refractivity retrievals) to read as follows:

*GNSS RO inversion also relies on an assumption of local spherical symmetry --- that atmospheric temperature, pressure, and water vapor depend only on height in the vicinity of*

*an RO sounding --- for retrieval, but this assumption is strongly violated by water vapor morphology in the PBL, especially in low latitudes.*

- P3L64: "These new ..." So far, several low SNR missions, e.g. like Spire, have also shown good penetration into the PBL. I think it is not as much a penetration problem, but rather a problem of how much of the bending to very low SLTA values is captured by the RO instrument. Without capturing this deep signal, the penetration can still get into the PBL, but the retrieved profiles are more biased as they miss this deep part.

We thank the referee and agree with this comment agree. However, deep signals are easier to capture with higher SNR data. We have clarified this to read

*These new, exceptionally large SNRs allow deeper signal tracking that ever before, even in the presence of extreme bending and super-refraction, reducing refractivity biases.*

- P4L95: The cut off can also be performed in time (looking at SRN), or SLTA. I believe this is the more common approach, so that the WO algorithm is not operating on all data. But maybe I misunderstood your statement.

We have added the following information about quality control between the three centers.

*ROM SAF quality control screens out retrieving with impact parameter noise above a certain threshold, as well as rejecting retrievals with bending angle or refractivity values sufficiently divergent from ionospheric correction and climate models. UCAR similar screens out retrievals with bending angle or refractivity values that sufficiently diverge from climate models, as well as bending angle or refractivity values with a high standard deviation of relative difference with the climate model. UCAR also rejects occultations with sufficiently high differences in bending angle between the L1 and L2 channels. JPL quality control rejects occultations with negative bending angles and high-altitude bending angles sufficiently divergent from a fit that is exponential with height. Similar to ROM SAF and UCAR, JPL also rejects retrievals with refractivity values sufficiently divergent from a climatology model at any height. JPL is distinct from the other two centers in that it also screens retrievals with temperatures sufficiently different from the climate model. For more a more detailed comparison of quality control between the three processing centers, see Ho et al. 2012 and Sokolovskiy 2020.*

- Section 2: I don't think, we need to go through the basics of RO here. It has been published often enough. Section 2 can thus be shortened to focus on the relevant.

We have shortened the paragraphs describing the background of RO here, while retaining a barebones summary of processing steps relevant to retrieval differences.

- P5L133: Why is C2 here 1500V/V, and above 2000V/V?

We thank the referee for catching this typo. We have corrected this to 1500 V/V.

- P7L194: My last comment on UCAR reference location - maybe mention it here, as this is easily over read or misunderstood where the ERA5 data actually came from.

We have added a note explaining this to the end of this section. See response to the last comment.

- Eq 3: This is the rather old, and by now slightly modified equation for refractivity. Did you really use this, or did you use more recent versions, as e.g. also used in the ROPP tool?

Smith-Weintruab relation given in the paper is the version we used. While we acknowledge that newer revisions to this refractivity computation are common, this form is still used regularly. See, e.g., these papers from the last 6 months in AMT:

Xu, X., Han, W., Wang, J., Gao, Z., Li, F., Cheng, Y., and Fu, N.: Quality assessment of YUNYAO radio occultation data in the neutral atmosphere, Atmos. Meas. Tech., 18, 1339–1353, https://doi.org/10.5194/amt-18-1339-2025, 2025.

Katona, J. E., de la Torre Juárez, M., Kubar, T. L., Turk, F. J., Wang, K.-N., and Padullés, R.: Cluster analysis of vertical polarimetric radio occultation profiles and corresponding liquid and ice water paths from Global Precipitation Measurement (GPM) microwave data, Atmos. Meas. Tech., 18, 953–970, https://doi.org/10.5194/amt-18-953-2025, 2025.

Winning Jr., T. E., Xie, F., and Nelson, K. J.: Assessing the ducting phenomenon and its potential impact on Global Navigation Satellite System (GNSS) radio occultation refractivity retrievals over the northeast Pacific Ocean using radiosondes and global reanalysis, Atmos. Meas. Tech., 17, 6851–6863, https://doi.org/10.5194/amt-17-6851-2024, 2024.

- Figure 1: Is there a possibility to use more distinctive colors? Not sure I know what lime green is vs. sage green. You could even leave the centre white. On second thought, I don't see the need for this Venn diagram, a table would also

do. There, you could even provide further info, like total number of occs, setting vs rising, etc.

We thank the referee for these comments. have changed the colors to be more distinctive. We note that the program used to make the Venn diagram is designed so that the color of each segment is a combination of the overlapping circles in that area (e.g. the segment of the Venn diagram where red and yellow overlap would be orange). This makes the Venn diagram easier to interpret intuitively, but prevents the colors from being highly distinctive. We have increased the width of the lines separating the segments so that different segments can easily be identified even by colorblind readers.

We find that the Venn diagram is easier to understand at a quick glance than a table. However, we have also added a table next to the Venn diagram for more detailed information.

P8L223: The found differences in occultations processed might be due to which version is available in the AWS repository. I think, you need to provide some more info there, as e.g. the ROM SAF data might come from their last reprocessing, which used an older UCAR atmPhs data set, while the UCAR data might already be the most recent (I believe 2021) reprocessing.

Also, more generally, processing of RO data improves continuously, thus differences found here could be due to when the processing system was actually set up. The ROM SAF is going to release an updated COSMIC-1 reprocessing in the near future, which might than be more similar to what UCAR did.

The versions hosted on the AWS repository are the 2019 CDR processing for ROM SAF, the 2021 UCAR reprocessing, and JPL's version 2.6. We have added this information to the Model and data section. Which reads

*The retrieval versions used were  the 2021 UCAR reprocessing (UCAR 2021), and JPL's reprocessing version 2.6, and the 2019 processing in the Climate Data Repository (CDR) from ROM SAF (ROMSAF 2021).*

We have also added a note in the quality control section which now reads

*ROM SAF uses UCAR calibrated phase data for their retrievals, which may explain why they share a higher percentage of retrievals not processed by JPL (15.4\%). This may indicate a difference in quality control between UCAR and JPL at the calibrated phase level. However, we note that the calibrated phase files used by the ROM SAF retrievals*

*hosted on the AWS Registry of Open Data are an older reprocessing than those used by the UCAR retrievals.*

We agree that different reprocessings will change the inter-center retrieval differences. Our intention is to characterize differences in current retrieval algorithms, to set the stage for future work understanding what aspects of the retrieval algorithms cause these differences. This work will apply to any future reprocessings, as well.

P8L226: Are these 2% from one particular GPS satellite? Or in any other way having a common characteristic?

This is an interesting idea. However, we have found no common characteristic. We searched for shared characteristics in the satellite producing the occultation, the receiving satellite, date of occultation, geographic distribution, and whether the occultation was rising or setting and found no correlation.

- P8L224: remove at?

Thank you, we have removed this typo.

- Figure 4: when zooming in, it seems that pixels overlap with land. I assume this is only the plotting, correct? So if the pixel has even only a small ocean part, it is plotted, but only the ocean occs are included? How do you account for orography near the coast? And, given that the ocean part of that pixel is rather small, you'd also have only few occultations that count towards the median - thus, these pixels could have higher noise values.

Yes, the only occultations contained within the bins/pixels for this (and all of the map-like figures in this work) are over the ocean. The bins are set to be 5x5 degrees, so a 5x5 degree bin along a coastline will appear to overlap with land, but the color of the bin will correspond to the mean/median of only the ocean occultations. We agree that this means that the bins closer to the coast will typically container fewer occultations and thus be more susceptible to noise. In the figures later in the paper, we apply a statistical mask to account for this. In the median height figure, however, we were particularly interested in looking for finer scale zonal and meridional trends in minimum height – e.g. a potential band of low better penetration along the ITCZ. We acknowledge that this does mean cells with few

occultations, such as those that overlap with a coastline, are especially susceptible to noise and the anomalous high or low minimum height values in a single cell should not be taken to be indicative of a true pattern. Rather, several many adjacent cells that all share a low or high minimum height value (e.g. the strings of higher minimum height/greener cells along both the northern and southern edges of the Tropics) are more indicative of a real physical trend.

This figure has been moved to the appendix; however, we have added the following note to the caption of the first binned map which appears in the paper.

*Note that bins which overlap the coastline still represent the median of only occultations over ocean. Many of these bins therefore contain fewer samples and are more susceptible to noise than bins whose areas are entirely over the ocean.*

The original figure, now in the appendix, uses no statistical mask. In this figure caption, we have further clarify the influence of noise on the bins, we have added the following.

*No statistical mask is used in this figure so that fine, small-scale zonal and meridional patterns in minimum height are shown, but abnormally high or low minimum height values in individual cells --- especially those along coastlines --- may be due to larger noise.*

- P11L273: no need to hypothesize here, that has been shown various times, maybe just include an appropriate reference.

We have added a reference to

Xie, F., Wu, D., Ao, C., Kursinski, E., Mannucci, A., and Syndergaard, S.: Super-refraction effects on GPS radio occultation refractivity in marine boundary layers, Geophys. Res. Lett., 37, L11 805, https://doi.org/10.1029/2010GL043299, 2010.

This section now reads

*It has been shown that subsidence produces sharp refractivity gradients that result in a bias towards higher minimum heights (poorer penetration) \citep{XieWuAo2012}, explaining the green bands of weaker penetration along the outer edges of the Tropics. We hypothesize that convection along the ITCZ may reduce the vertical discontinuities in*

*refractivity, causing quality control measures to allow processing of occultations deeper into the PBL.*

- P11L280: Interpolation linear in log space? As REF varies essentially like pressure, exponentially?

The interpolation is linear in geopotential height. We have clarified this text to read

*We compare both JPL and ROM SAF to UCAR, using an interpolation (without extrapolation) linear in geopotential height on a 100-m isohypsic grid between 0 and 20 km to compare occultations at different processing centers*

- P13L297: Did you also apply this minimum for the Figure 4 map?

No; we were hoping to identify finer structure zonal and meridional patterns in the Fig. 4 map (e.g. better penetration along the ITCZ, worse penetration along the edges of the Hadley cell, impacts of the Gulf Stream or Pacific cold tongue, etc.). In Fig. 6 we were hoping to identify more statistically robust differences between processing centers. The text added to the Fig. 4 caption should help clarify this.

*Maps of median minimum height on 5°bins for maritime occultations shared by all three processing centers for DJF (left) and JJA (right) 2008. Yellow coloring indicates more conservative (higher) minimum height. The minimum heights for all three centers are highest in the Tropics, especially for JPL, confirming the effects seen in Fig. 2. Bins with a seasonal mean precipitation of at least 5 mm $d-1$ on average are labeled with red stippling. Note that bins which overlap the coastline still represent the median of only occultations over ocean. Many of these bins therefore contain fewer samples and are more susceptible to noise than bins whose areas are entirely over the ocean. No statistical mask is used in this figure so that fine, small-scale zonal and meridional patterns in minimum height are shown, but abnormally high or low minimum height values in individual cells — especially those along coastlines — may be due to larger noise.*

- P15L320: BTW, do you do any sea ice screening? Could reflections, entering the retrieval scheme, cause these differences?

This is an intriguing hypothesis. We did not perform sea ice screening. Cardellach et al. 2008 tentatively found that sea ice reflection does not significantly impact retrievals. Cardellach and Oliveras 2016 found that RO retrievals that feature reflections (from sea ice or other ocean reflections) improve the negative refractivity bias (*not* causing a positive bias); however, they find that this effect is stronger at mid-latitudes than polar regions,

which does not correspond to the positive bias in our results. We have added the following text to reflect this:

*Furthermore, while these regions may feature reflections from sea ice, Cardellachet al. 2008 demonstrated that these refrlections do not noticeably bias RO retrievals.  iveras2016 showed that retrievals that feature reflection (from sea ice or from other sources) have lower negative refractivity biases, but these impacts are strongest in mid- and low-latitudes --- the opposite of the regions shown in Fig. 6 --- and do not induce a positive bias.*

Cardellach, E. and Oliveras, S.: Assessment of a potential reflection flag product, Tech. rep., IEEC, Barcelona, Spain, https://rom-saf.430eumetsat.int/general-documents/rsr/rsr_23.pdf, 2016.

Cardellach, E., Oliveras, S., and Rius, A.: Applications of the Reflected Signals Found in GNSS Radio Occultation Events, https://www.ecmwf.int/sites/default/files/elibrary/2008/7460-applications-reflected-signals-found-gnss-radio-occultation-events.pdf, 2008.

- Section 3.5: I am unsure why ENSO should have an impact, globally. Any reason for looking into this separately? And, maybe just mention in a sentence the non existing ENSO impact, instead of adding a figure that reveals no info.

This is a good point. The impact of ENSO would be local, not global. Using a global map allows us to demonstrate the localized impact of ENSO (or lack thereof). We have removed this section and figure from the text, instead replacing it with two sentences at the end of the "Comparison with models" section, which reads

*We also considered the effect of the El Niño Southern Oscillation (ENSO) on refractivity biases. The strongest El Niño covered by COSMIC-1 was the 2015 event, which peaked in strength in the autumn of that year. Comparison of the refractivity bias of UCAR relative to ERA5 for September, October, and November (SON) of both 2008 and 2015 showed no noticeable difference. We thus conclude that ENSO does not have a significant impact on the refractivity bias.*

- P16L345: … is hightest *in*? the subsiding…

We thank the referee and have corrected this typo.

- P18L370: … the the …

We thank the referee and have corrected this typo.

- P18L387: You used the same ERA5 profiles for all occultations? An issue here is that UCAR uses a different reference tangent point than e.g. the ROM SAF. So the ERA5 UCAR profiles could be more than 100km off to where ROM SAF puts the reference - but at least this is consistent in your use (we right now have a C2 discussion where some of the biases found are due to the different reference tangent points). Maybe mention that higher up.

I don't believe this introduces an inter-center bias in our results. UCAR and ROM SAF do define their tangent point locations differently. However, for each occultation, the ROM SAF, UCAR, and JPL retrievals are all compared to the same ERA5 profile (chosen to be the one closest to the UCAR reference tangent point). Since the three RO retrievals are still the same occultation (they are collocated) – just with different definition of tangent point – they should all be compared to the same ERA5 reference. Choosing the ERA5 profile to match the ROM SAF definition of reference would change the bias of the processing centers relative to ERA5, but not the bias between the processing centers. We have added a note to the bottom of the Data and model section explaining this, which reads

*We note a source of error in the RO-ERA5 comparisons: Each processing center defines the location of their reference tangent point differently. The ERA5 profile selected for each occultation was chosen to be the profile closest to the UCAR reference tangent point. Since the retrievals for each center remain fundamentally the same occultation --- the differing tangent points are due only to differing tangent point definitions --- they should all be compared to the same ERA5 profile. Using the ERA5 profile closest to, say, the ROM SAF definition of the reference tangent point may alter the RO-ERA5 bias, but should not change the bias between the processing centers.*